# Uncertainty-aware Guided Diffusion for Missing Data in Sequential Recommendation

## Abstract

Denoising diffusion models (DDMs) have shown significant potential in generating oracle items that best match user preference with guidance from user historical interaction sequences. However, the quality of guidance is often compromised by the unpredictable missing data in the observed sequence, leading to suboptimal item generation. To tackle this challenge, we propose a novel uncertainty-aware guided diffusion model (DreamMiss) to alleviate the influence of missing data. The core of DreamMiss is the utilization of a dual-side Thompson sampling (DTS) strategy, which simulates the stochastical mechanism of missing data without disrupting preference evolution. Specifically, we first define dual-side probability models to capture user preference evolution, taking into account both local item continuity and global sequence stability. We then strategically remove items based on these two models with DTS, creating uncertainty-aware guidance for DDMs to generate oracle items. This can achieve DDMs' consistency regularization, enabling them to resist missing data in real scenarios. Additionally, to accelerate sampling in the reverse process, DreamMiss is implemented under the framework of denoising diffusion implicit models (DDIM). Extensive experimental results show that DreamMiss significantly outperforms baselines in sequential recommendation.

## 1 Introduction

Sequential Recommendation (Kang & McAuley, 2018; Xie et al., 2022) is to predict the next item that aligns with a user's preferences based on his/her historical interaction sequence. Unlike conventional studies (Xie et al., 2022; Qiu et al., 2022; Hidasi et al., 2016) that learn to classify target items from sampled negatives, recent studies (Sachdeva et al., 2019; Xie et al., 2021; Rajput et al., 2023) shift towards generating oracle items (Yang et al., 2023) with generative models that best match user preference and grounding them in target items. A promising direction is employing denoising diffusion models (DDMs) (Yang et al., 2023; Li et al., 2024; Niu et al., 2024), which add noise to the next items and iteratively denoise them toward oracle items, guided by interaction history conditions.

However, we argue that DDMs' ability to generate oracle items is largely constrained by the quality of guidance (*i.e.,* the representation of interaction sequence). Typically, user interaction histories are only partially observed, with missing data occurring unpredictably (Zheng et al., 2022). Consider the case of item browsing as illustrated in Figure 1. The recommender system might only observe a partial sequence, with items $A$, $B$, and $C$ missing due to various factors, such as privacy concerns (Feng et al., 2023) or technical limitations (Xu et al., 2020). Consequently, DDMs may be misled by the unreliable guidance signal from the observed sequence and generate suboptimal oracle items. Usually, data missing is **uncertain** in the observed sequence (Wang et al., 2023a; Fan et al., 2022b), as it is hard to infer where the missing occurs and what content it might be due to the invisibility of complete sequence. Thus, leading approaches that aim to tackle missing data issues, such as imputation-based algorithms (which fill in the missing data) (Steck, 2013; Hikmawati et al., 2024) and latent variable decomposition methods (which remove the latent variables causing missing data, such as exposure bias) (Saito et al., 2020; Li et al., 2023b), may introduce additional errors or distort user preference accidentally, as shown in Figure 1.

To enhance the robustness of DDMs, rather than inferring the occurred missing data, we aim to simulate the uncertain data missing, creating uncertainty-aware guidance and achieving DDMs' consistency regularization (Zhang et al., 2020). To simulate data missing stochastically while

Figure 1: Phenomenon of uncertain data missing in sequence and the comparison between methods to address it. The green curve represents the evolution of user preference over time. The interaction sequences inside the blue boxes serve as the guidance for diffusion models while the next item "bag" serves as the input of DDMs.

maintaining users' underlying preference evolution pattern, we introduce the Dual-side Thompson Sampling (Russo et al., 2018) (DTS) which samples by exploiting known user preference. Specifically, DTS hires two probability models — one operating at the local item side and the other at the global sequence side — to capture the dynamic preference:

- The local model depicts the continuity between adjacent items, reflecting shifts in user preference. As depicted in Figure 1, a high continuity score of 0.9 indicates a coherent preference for clothes, while a low score of 0.2 suggests a preference shift from mobile phones to headphones.

- The global model evaluates the stability of each entire sequence, by calculating the entropy of continuity score distribution — a high stability score indicates a stable preference, whereas a low score reflects a volatile preference. For instance, the sequence in Figure 1 experienced two significant fluctuations, resulting in a stability score of 0.6.

High-continuity items in high-stability sequences are more likely to be removed by the dual-side Thompson sampling strategy, which has little impact on preference evolution. This is evidenced by the consistent green curves between the observed sequence and the one edited by our method, as illustrated in Figure 1. Then, we treat such stochastically edited sequences as uncertainty-aware guidance for DDMs to generate the oracle items. Incorporating uncertainty into the guidance can achieve consistency regularization (Zhang et al., 2020) which endows DDMs with insensitivity to preference-preserving perturbations (*i.e.,* simulated missing data) and can extrapolate to resist original missing data in real scenarios. We term this uncertainty-aware guided diffusion model as DreamMiss, designed to mitigate the influence of missing data in sequential recommendation. To further improve the efficiency, we utilize denoising diffusion implicit models (DDIM) to generate oracles rather than denoising diffusion probabilistic models (DDPM), which can accelerate the sampling in the reverse process. We validate the effectiveness of DreamMiss with experiments on three datasets. Extensive experimental results demonstrate that DreamMiss can improve the performance of sequential recommendation, outperforming multiple leading models.

## 2 PRELIMINARIES

In this section, we first detail denoising diffusion implicit models (DDIM) (Song et al., 2021), highlighting the ability to accelerate the sampling process. We then introduce Thompson sampling. Finally, we formulate the task of generative sequential recommendation with diffusion models.

### 2.1 DENOISING DIFFUSION IMPLICIT MODELS (DDIM)

DDIM (Song et al., 2021) is a generative model designed to learn data distribution and generate high-quality samples with accelerated sampling than the original DDPM (Ho et al., 2020). Here we elaborate on the forward and reverse processes of DDIM.

**Forward process:** Compared with DDPM, the forward process of DDIM is not restricted as a Markovian chain. Given an input data sample $\mathbf{x}^0 \sim q(\mathbf{x}^0)$, the forward diffusion process can be defined as: $q(\mathbf{x}^t|\mathbf{x}^0) = \mathcal{N}(\mathbf{x}^t; \sqrt{\alpha_t}\mathbf{x}^0, (1-\alpha_t)\mathbf{I})$, where $t \in [1, \ldots, T]$ represents the diffusion step, $[\alpha_1, \ldots, \alpha_T]$ denotes a variance schedule. We can have: $\mathbf{x}^t = \sqrt{\alpha_t}\mathbf{x}^0 + \sqrt{1-\alpha_t}\boldsymbol{\epsilon}$.

**Reverse process:** Given $\mathbf{x}^T \sim \mathcal{N}(\mathbf{0}, \mathbf{I})$, DDIM eliminates the noises to recover $\mathbf{x}^0$ step by step. Formally, the reverse process of DDIM from $\mathbf{x}^t$ to $\mathbf{x}^{t-1}$ is:

$$p_{\theta,\sigma}(\mathbf{x}^{t-1}|\mathbf{x}^t) = \mathcal{N}\left(\sqrt{\alpha_{t-1}}\,\hat{\mathbf{x}}^0 + \sqrt{1-\alpha_{t-1}-\sigma_t^2}\frac{\mathbf{x}^t - \sqrt{\alpha_t}\,\hat{\mathbf{x}}^0}{\sqrt{1-\alpha_t}}, \sigma_t^2 I\right), \tag{1}$$

$$\hat{\mathbf{x}}^0 = (\mathbf{x}^t - \sqrt{1-\alpha_t}\boldsymbol{\epsilon}_\theta(\mathbf{x}^t))/\sqrt{\alpha_t}. \tag{2}$$

When the standard variance $\sigma_t = \eta\sqrt{\frac{1-\alpha_{t-1}}{1-\alpha_t}}\sqrt{1-\frac{\alpha_t}{\alpha_{t-1}}}$, the reverse process is equivalent to DDPM. When $\sigma_t = 0$, it becomes deterministic DDIM. $\boldsymbol{\epsilon}_\theta$ denotes the denoising neural network (*e.g.,* U-Net (Wang et al., 2023d) or Transformer (Peebles & Xie, 2023)) parameterized by $\theta$, which is trained to approximate the data distribution $q\left(\mathbf{x}^0\right)$ by maximizing the evidence lower bound of the log-likelihood $\log p_\theta\left(\mathbf{x}^0\right)$. The training loss can be derivated as follows (Song et al., 2021):

$$\mathcal{L} = \sum_{t=1}^{T} \frac{1}{2d\sigma_t^2\alpha_t} \mathbb{E}_{\mathbf{x}^0,\boldsymbol{\epsilon}}\left[\left\|\boldsymbol{\epsilon}_\theta(\sqrt{\alpha_t}\mathbf{x}^0 + \sqrt{1-\alpha_t}\boldsymbol{\epsilon}, t) - \boldsymbol{\epsilon}\right\|_2^2\right], \tag{3}$$

where $\boldsymbol{\epsilon}_\theta(\sqrt{\alpha_t}\mathbf{x}^0 + \sqrt{1-\alpha_t}\boldsymbol{\epsilon}, t)$ is the output of the denoising network to predict the noises $\boldsymbol{\epsilon}$ that add in the forward process, $d$ is the dimension of $\mathbf{x}^0$.

**Accelerated sampling:** Since the denoising objective $\mathcal{L}$ is independent of a specific forward process as long as $q(\mathbf{x}^t|\mathbf{x}^0)$ is fixed (Song et al., 2021), we can redefine the non-Markovian forward process with a subsequence $[\tau_1, \tau_2, \cdots, \tau_S]$ from $[1, \ldots, T]$ as: $q(\mathbf{x}^{\tau_s}|\mathbf{x}^0) = \mathcal{N}(\mathbf{x}^{\tau_s}; \sqrt{\alpha_{\tau_s}}\mathbf{x}^0, (1-\alpha_{\tau_s})\mathbf{I})$. Then the reverse process can be reformulated as:

$$\mathbf{x}^{\tau_{s-1}} = \sqrt{\alpha_{\tau_{s-1}}}\left(\frac{\mathbf{x}^{\tau_s} - \sqrt{1-\alpha_{\tau_s}}\boldsymbol{\epsilon}_\theta(\mathbf{x}^{\tau_s}, \tau_s)}{\sqrt{\alpha_{\tau_s}}}\right) + \sqrt{1-\alpha_{\tau_{s-1}}-\sigma_{\tau_s}^2}\boldsymbol{\epsilon}_\theta(\mathbf{x}^{\tau_s}, \tau_s) + \sigma_{\tau_s}\boldsymbol{\epsilon}. \tag{4}$$

With a smaller number of steps $S$ compared to the original $T$, the sampling phase can be accelerated. A More detailed comparison of the rationale of DDIM and DDPM is presented in the Appendix.

## 2.2 THOMPSON SAMPLING

Thompson sampling (TS) (Russo et al., 2018) has emerged as a prominent exploration strategy for decision-making under uncertainty (McDonald et al., 2023; Wang & Zhou, 2020). To achieve a balance between exploration and exploitation (Osband & Roy, 2015), TS utilizes a probability model to sample greedily based on the values of execution results from the last round. Specifically, given the value $v$, the probability model of TS can be parameterized as $F(v, p)$, where $p$ is a random variable that ranges from 0 to 1. A higher value of $v$ can result in a higher sampling probability $\hat{p}$ from the probability model. Formally, at each round, we have the sampling probability:

$$\hat{p} \sim F(v, p). \tag{5}$$

Then TS executes based on the sampling probability $\hat{p}$, updating the value $v$ and probability model $F(v, p)$ based on the execution results.

## 2.3 TASK FORMULATION

For generative sequential recommendation, the goal is to generate the next item tailored to the target user conditioned on their historical interaction sequence. The mainstream solutions to this task are from the embedding perspective. Formally, we denote a user's historical interaction sequence as $\mathbf{e}_{1:N-1} = [\mathbf{e}_1, \mathbf{e}_2, \ldots, \mathbf{e}_{N-1}]$, where $\mathbf{e}_n$ represents the embedding of the $n$-th item the user has interacted with in chronological order. The subsequent item of this sequence, which we aim to generate, is represented as $\mathbf{e}_N$. To apply DDMs in generative recommendation, following prior studies (Yang et al., 2023; Niu et al., 2024), noise is first added to $\mathbf{e}_N^0$ (equivalent to $\mathbf{e}_N$), followed

Figure 2: The overview of our proposed DreamMiss framework, which can create uncertainty-aware guidance for DDMs to mitigate the influence of missing data in sequential recommendation.

by a denoising process leveraging the guidance signal $\mathbf{g}$ extracted from interaction history $\mathbf{e}_{1:N-1}$ to ensure the generated oracle items align closely with user preferences. At its core is to model the item generation distribution $p_\theta(\mathbf{e}_N^{t-1}|\mathbf{e}_N^t, \mathbf{g})$ at each $t$-th denoising step, and inference step by step to generate the oracle items $\mathbf{e}_N^0$.

## 3 METHODOLOGY

In this section, we present our proposed DreamMiss, designed to mitigate the impact of missing data on DDMs, as shown in Figure 2. We begin by detailing the uncertainty-aware guidance creation process in Section 3.1, which employs a dual-side Thompson sampling strategy to simulate missing data in user behaviors. Next, we describe the learning and generating phases of DreamMiss in Section 3.2. Finally, we explain the consistency regularization of DreamMiss in Section 3.3.

### 3.1 UNCERTAINTY-AWARE GUIDANCE WITH DUAL-SIDE THOMPSON SAMPLING

In real-world scenarios, missing data is inherently uncertain and unpredictable(Wang et al., 2023a; Fan et al., 2022b), so we simulate it stochastically rather than recovering it, creating uncertainty-aware guidance and realizing DDMs' consistency regularization. To preserve user preference evolution during the simulation, we introduce a Dual-side Thompson Sampling (DTS) strategy, sampling and removing items by exploiting user preference evolution with two probability models — one at the local item level and the other at the global sequence level.

#### 3.1.1 DEFINITION OF TWO PROBABILITY MODELS

User preferences often exhibit dynamic shifts between items within an interaction sequence. To capture these preference shifts locally, we introduce the concept of *continuity scores* to measure the similarity between adjacent items. Formally, given a sequence $\mathbf{e}_{1:N-1} = [\mathbf{e}_1, \mathbf{e}_2, \ldots, \mathbf{e}_{N-1}]$, the continuity score for each item $\mathbf{e}_n$ within the sequence is defined as:

$$\text{con}_n = \frac{\exp(\text{sim}(\mathbf{e}_n, \mathbf{e}_{n+1}))}{\sum_{n'=1}^{N-2} \exp(\text{sim}(\mathbf{e}_{n'}, \mathbf{e}_{n'+1}))}, \quad n = 1, 2, \ldots, N-2, \tag{6}$$

where $\text{sim}(\cdot, \cdot)$ represents the cosine similarity function, $\text{con}_n$ is the continuity score normalized using the *softmax* function. Intuitively, a higher continuity score indicates a greater similarity between temporally adjacent items, indicating a stronger level of shared preference.

In addition to local preference shifts, user preferences often fluctuate throughout the entire sequence (Li et al., 2018). To assess the degree of these preference fluctuations, we calculate the entropy value $h$ for each sequence with the continuity scores within it as below:

$$h = -\sum_{n=1}^{N-2} \text{con}_n \log(\text{con}_n). \tag{7}$$

Then we define the *stability score* $\text{sta}_k$ for each sequence by normalizing the sequence entropy value $h_k$ in a batch with a softmax function.

$$\text{sta}_k = \frac{\exp(h_k)}{\sum_{k'=1}^{K} \exp(h_{k'})}, \quad k = 1, 2, \ldots, K, \tag{8}$$

where $K$ is the number of sequences in the batch. A higher stability score reflects a higher entropy value, suggesting that user preferences remain largely unchanged throughout the interaction sequence.

We then parameterize the two probability models with value $\text{con}_n$ and $\text{sta}_k$ as introduced in Section 2.2. Formally, we define the local item-side probability model as $L(\text{con}_n, p_n)$ and the global sequence-side model as $G(\text{sta}_k, p_k)$, where $p_n$ and $p_k$ are the random variables ranging from 0 to 1. Higher values in $\text{con}_n$ and $\text{sta}_k$ lead to higher sampling probabilities from their respective models.

### 3.1.2 STOCHASTIC EDITING WITH DUAL-SIDE THOMPSON SAMPLING

To simulate missing data while maintaining user preference, the DTS samples sequences to edit and items to remove based on the two probability models defined in Section 3.1.1. Formally, we have:

$$\hat{p}_n \sim L(\text{con}_n, p_n), \quad \hat{p}_k \sim G(\text{sta}_k, p_k), \tag{9}$$

where $\hat{p}_k$ is the sampling probabilities of the $k$-th sequence $\mathbf{e}_{1:N-1}$ to be edited, and $\hat{p}_n$ is probability for the $n$-th item $\mathbf{e}_n (1 \leq n \leq N - 2)$ within sequence $\mathbf{e}_{1:N-1}$ to be discarded. To preserve the last item $\mathbf{e}_{N-1}$ in the sequence, we manually set $\hat{p}_{N-1} = 0$. Since $\text{con}_n$ and $\text{sta}_k$ represent the local continuity and global stability respectively, the dual-side Thompson sampling strategy tends to sample items with higher continuity in sequences with greater stability scores to remove. Removing such data is expected to have little impact on the original preference shifts, as shown in the consistent green curves in Figure 2. Therefore, we can simulate uncertain data missing, while preserving the underlying preference evolution pattern.

Having established $\hat{p}_k$ and $\hat{p}_n$, we can decide whether the $n$-th item in the $k$-th sequence would be discarded. Formally, the stochastically edited sequence is obtained as:

$$\mathbf{e}'_{1:N-1} = \begin{cases} [\,\text{edit}(\mathbf{e}_n)\,]_{n=1}^{N-1} & \text{if } 1 - \hat{p}_k < \lambda_1 \\ \mathbf{e}_{1:N-1} & \text{otherwise} \end{cases}, \quad \text{edit}(\mathbf{e}_n) = \begin{cases} \Phi & \text{if } 1 - \hat{p}_n < \lambda_2 \\ \mathbf{e}_n & \text{otherwise} \end{cases}, \tag{10}$$

where $\lambda_1$ and $\lambda_2$ are thresholds for sampling probabilities $\hat{p}_k$ and $\hat{p}_n$, ranging from 0 to 1, which control the proportion of removed items. Higher values of $\lambda_1$ and $\lambda_2$ result in more items in more sequences being removed.

We then encode the probabilistically edited sequence $\mathbf{e}'_{1:N-1}$ using a Transformer encoder T-enc to obtain the uncertain-aware guidance $\mathbf{g}$:

$$\mathbf{g} = \text{T-enc}(\mathbf{e}'_{1:N-1}). \tag{11}$$

In this way, the uncertainty-aware guidance of the interaction history is established by the dual-side Thompson sampling strategy, simulating missing data stochastically while preserving the evolution of users' dynamic preferences.

### 3.2 DIFFUSION MODEL FOR RECOMMENDATION

Having acquired the uncertainty-aware guidance $\mathbf{g}$ as described in Section 3.1, we could leverage $\mathbf{g}$ to guide the diffusion model, enabling DreamMiss to recommend items robustly in the presence of missing data. To accelerate the inference, we employ DDIM introduced in Section 2.1 for DreamMiss to generate oracle items. Below, we detail DreamMiss's training and generating phases.

### 3.2.1 TRAINING PHASE

For joint training of both conditional and unconditional models, we train DreamMiss under the classifier-free guidance paradigm (Ho & Salimans, 2022). Specifically, we randomly replace the guidance $\mathbf{g}$ with a dummy token $\Phi$ with probability $\rho$, while keeping the others unchanged. We view the next item $\mathbf{e}_N$ as the input $\mathbf{e}_N^0$ and add noise to it, following: $\mathbf{e}_N^{\tau_s} = \sqrt{\alpha_{\tau_s}} \mathbf{e}_N^0 + \sqrt{1 - \alpha_{\tau_s}} \boldsymbol{\epsilon}$.

Similar to DreamRec (Yang et al., 2023), we employ an MLP as the denoising neural network $f_\theta(\cdot, \cdot, \cdot)$ to directly predict $\mathbf{e}_N^{\tau_s}$ into $\hat{\mathbf{e}}_N^0$, rather than the noise $\boldsymbol{\epsilon}$, guided by $\mathbf{g}$:

$$\hat{\mathbf{e}}_N^0 = f_\theta(\sqrt{\alpha_{\tau_s}}\mathbf{e}_N^0 + \sqrt{1-\alpha_{\tau_s}}\boldsymbol{\epsilon}, \mathbf{g}, \tau_s), \tag{12}$$

where $\hat{\mathbf{e}}_N^0$ denotes the prediction of $\mathbf{e}_N^0$. According to Equation 3, the loss function of DreamMiss can be formulated as:

$$\mathcal{L} = \sum_{s=1}^{S} \frac{1}{2d\sigma_{\tau_s}^2(1-\alpha_{\tau_s})}\mathbb{E}_{\mathbf{e}_N^0, \boldsymbol{\epsilon}}\left[\|\hat{\mathbf{e}}_N^0 - \mathbf{e}_N^0\|_2^2\right]. \tag{13}$$

### 3.2.2 GENERATING PHASE

Having trained the denoising model $f_\theta(\cdot, \cdot, \cdot)$, DreamMiss can generate the oracle items step by step. Specifically, to integrate the conditional and unconditional generation under the classifier-free guidance paradigm, the denoising function is modified with a linear combination:

$$\tilde{f}_\theta(\mathbf{e}_N^{\tau_s}, \mathbf{g}, \tau_s) = (1+w)f_\theta(\mathbf{e}_N^{\tau_s}, \mathbf{g}, \tau_s) - wf_\theta(\mathbf{e}_N^{\tau_s}, \Phi, \tau_s), \tag{14}$$

where the hyperparameter $w$ controls the guidance strength. A high value of $w$ increases reliance on the guidance $\mathbf{g}$, but it may lead to overfitting. Following Equation 4, the reverse denoising step from $\tau_s$ to $\tau_{s-1}$ can be expressed as:

$$\mathbf{e}_N^{\tau_{s-1}} = \sqrt{\alpha_{\tau_{s-1}}}\tilde{f}_\theta(\mathbf{e}_N^{\tau_s}, \mathbf{g}, \tau_s) + \sqrt{1-\alpha_{\tau_{s-1}}}\frac{\mathbf{e}_N^{\tau_s} - \sqrt{\alpha_{\tau_s}}\tilde{f}_\theta(\mathbf{e}_N^{\tau_s}, \mathbf{g}, \tau_s)}{\sqrt{1-\alpha_{\tau_s}}}. \tag{15}$$

For a user with interaction sequence encoded as $\mathbf{g}$, the oracel item $\mathbf{e}_N^0$ is generated by denosing a Gaussian sample $\mathbf{e}_N^{\tau_S} \sim \mathcal{N}(\mathbf{0}, \mathbf{I})$ for $\tau_S$ times with Equation 15. Once the oracle item is generated, we retrieve the K-nearest items from the candidate set to provide the top-K recommendation results. See Appendix D for the algorithms of training and generating phases of DreamMiss.

### 3.3 THEORETICAL ANALYSIS FOR SIMULATING MISSING DATA OF DREAMMISS

We denote the original missing data in sequence as $\delta$, the simulated missing data as $\delta'$, the unavailable complete sequence as $\mathbf{e}_{1:N-1} \oplus \delta$, the stochastically edited sequence $\mathbf{e}'_{1:N-1}$ as $\mathbf{e}_{1:N-1} \ominus \delta'$. We justify our method for addressing missing data issues through extrapolation and consistency regularization (Zhang et al., 2020; Tack et al., 2022). .

**Extrapolation** (Krueger et al., 2021): Let $\hat{\mathbf{g}}$, $\bar{\mathbf{g}}$, and $\tilde{\mathbf{g}}$ denote the guidance encoded from observed sequence $\mathbf{e}_{1:N-1}$, unavailable complete sequence $\mathbf{e}_{1:N-1} \oplus \delta$ and simulated sequence $\mathbf{e}_{1:N-1} \ominus \delta'$, respectively. We can derive the following inequality for extrapolation:

$$\|f_\theta(\mathbf{e}_N^{\tau_s}, \bar{\mathbf{g}}, \tau_s) - f_\theta(\mathbf{e}_N^{\tau_s}, \hat{\mathbf{g}}, \tau_s)\| \leq C\|f_\theta(\mathbf{e}_N^{\tau_s}, \hat{\mathbf{g}}, \tau_s) - f_\theta(\mathbf{e}_N^{\tau_s}, \tilde{\mathbf{g}}, \tau_s)\| \tag{16}$$

where $C$ is the constant. A detailed derivation process is presented in the Appendix B.3. According to Equation 16, enhancing DDMs' insensitivity to simulated missing data enables resilience against original missing data.

**Consistency regularization** (Zhang et al., 2020; Tack et al., 2022): According to Equation 10, the sequence $\mathbf{e}'_{1:N-1}$ are edited stochastically across different epochs. Thus $\hat{\mathbf{g}}$ and $\tilde{\mathbf{g}}$ can serve as the perturbated pairs, which share the same target $\mathbf{e}_N^0$. To ensure the insensitivity of DDMs to the simulated missing data, we leverage consistency regularization to minimize the distance between $f_\theta(\mathbf{e}_N^{\tau_s}, \hat{\mathbf{g}}, \tau_s)$ and $f_\theta(\mathbf{e}_N^{\tau_s}, \tilde{\mathbf{g}}, \tau_s)$. By completing the square, we have:

$$||f_\theta(\mathbf{e}_N^{\tau_s}, \hat{\mathbf{g}}, \tau_s) - f_\theta(\mathbf{e}_N^{\tau_s}, \tilde{\mathbf{g}}, \tau_s)||_2^2 \leq 2\left(||f_\theta(\mathbf{e}_N^{\tau_s}, \hat{\mathbf{g}}, \tau_s) - \mathbf{e}_N^0||_2^2 + ||f_\theta(\mathbf{e}_N^{\tau_s}, \tilde{\mathbf{g}}, \tau_s) - \mathbf{e}_N^0||_2^2\right). \tag{17}$$

Minimizing the right-hand side, which stems from our reconstruction loss, serves as an upper bound of the minimizer of the left-hand side, thus achieving consistency regularization. Such consistency regularization endows DDMs with insensitivity to the simulated missing data, allowing the extrapolation to resist the original missing data.

# 4 EXPERIMENTS

In this section, we conduct extensive experiments across three datasets to evaluate the effectiveness of DreamMiss by answering the following questions. **RQ1:** How does DreamMiss perform in the sequential recommendation compared with diverse baseline models? **RQ2:** How do different probability models bring improvements to our method? **RQ3:** How robust is DreamMiss to varying degrees of missing data in datasets? **RQ4:** How sensitive is DreamMiss to the strength of guidance and the thresholds of removing? **RQ5:** Can DreamMiss generalize on different sequence lengths?

## 4.1 EXPERIMENTAL SETTINGS

**Datasets.** We conduct experiments on three real-world datasets for sequential recommendation following the settings of DreamRec (Yang et al., 2023): YooChoose (Ben-Shimon et al., 2015), KuaiRec (Gao et al., 2022), and Zhihu (Hao et al., 2021). To mitigate cold-start issues, we implement a preprocessing step that excludes items with fewer than five interactions and sequences shorter than 3 interactions. The detailed statistics of the datasets are provided in Appendix E.1. For each dataset, we sort all sequences chronologically and split the data into training, validation, and testing sets in an 8:1:1 ratio, ensuring that later interactions don't leak into the training data (Ji et al., 2023). Additionally, to validate the effectiveness of DreamMiss on larger or diverse datasets from different domains, we conduct experiments on Steam, Amazon-beauty, and Amazon-toys in the appendix F.1.

**Baselines.** We compare DreamMiss against multiple leading approaches:

- Traditional sequential recommenders: GRU4Rec (Hidasi et al., 2016), Caser (Tang & Wang, 2018), SASRec (Kang & McAuley, 2018), Bert4Rec (Sun et al., 2019), and CL4SRec (Xie et al., 2022), which employ neural networks to capture user preferences.

- Latent variable decomposition methods: S-IPS (Wang et al., 2022) and AdaRanker (Fan et al., 2022a), which are designed to address the issues of selection or distribution bias.

- Generative recommenders: DiffRec (Wang et al., 2023c), DiffRIS (Niu et al., 2024), and DreamRec (Yang et al., 2023), which generate target items directly with DDMs.

- Imputation-based algorithms: DiffuASR (Liu et al., 2023a), CaDiRec (Cui et al., 2024), PDRec (Ma et al., 2024), STEAM (Lin et al., 2023), and SSDRec (Zhang et al., 2024). DiffASR, CaDiRec, and PDRec generate supplement items to the observed sequences with DDMs to enhance traditional sequential recommenders. STEAM and SSDRec aim to correct the interaction sequence by "insert" or other operations.

**Implementation Details.** Following DreamRec (Yang et al., 2023), we set the sequence length to 10, padding sequences with fewer than 10 interactions using a padding token. The parameter tuning strategy is detailed in the appendix E.2. We adopt the widely used metrics in sequential recommendation: hit ratio (HR@20) and normalized discounted cumulative gain (NDCG@20) (Kang & McAuley, 2018) to evaluate the recommendation performance. We report the average performance of five experimental runs, with their corresponding standard deviations.

## 4.2 MAIN RESULTS (RQ1)

To answer RQ1, we compare the recommendation performance of DreamMiss against multiple baselines. Table 1 presents the experimental results, which demonstrate that DreamMiss surpasses all baselines consistently across three datasets. For example, on the KuaiRec dataset, DreamMiss outperforms DreamRec, a generative recommender with DDMs that achieves second-best performance in most cases, with increases of $5.84\%$ and $13.84\%$ in HR@20 and NDCG@20, respectively. These indicate that incorporating uncertainty into guidance enhances the robustness of DDMs to unreliable sequences, thereby improving overall recommendation performance. Additionally, Dream-Miss achieves at least a $26.95\%$ higher NDCG@20 than methods specifically designed to handle missing data (*e.g.,* IPS, DiffuASR, and PDRec) on YooChoose, validating the advantages of the dual-side Thompson sampling strategy in addressing uncertain data missing. These results provide compelling evidence for the effectiveness of the proposed uncertainty-aware guided diffusion model in mitigating the impact posed by missing data in user behavior sequences, consequently enhancing recommendation performance.

Table 1: Overall performance of different methods for the sequential recommendation. The best score and the second-best score are bolded and underlined, respectively. The last row indicates the performance improvements of DreamMiss over the best-performing baseline method. H@20 and N@20 represent HR@20 and NDCG@20 respectively.

| Methods | YooChoose | | KuaiRec | | Zhihu | |
|---|---|---|---|---|---|---|
| | H@20(%) | N@20 (%) | H@20 (%) | N@20(%) | H@20(%) | N@20(%) |
| GRU4Rec | $3.89_{\pm0.11}$ | $1.62_{\pm0.02}$ | $3.32_{\pm0.11}$ | $1.23_{\pm0.08}$ | $1.78_{\pm0.12}$ | $0.67_{\pm0.03}$ |
| Caser | $4.06_{\pm0.12}$ | $1.88_{\pm0.09}$ | $2.88_{\pm0.19}$ | $1.07_{\pm0.07}$ | $1.57_{\pm0.05}$ | $0.59_{\pm0.01}$ |
| SASRec | $3.68_{\pm0.08}$ | $1.63_{\pm0.02}$ | $3.92_{\pm0.18}$ | $1.53_{\pm0.11}$ | $1.62_{\pm0.01}$ | $0.60_{\pm0.03}$ |
| Bert4Rec | $4.96_{\pm0.05}$ | $2.05_{\pm0.03}$ | $3.77_{\pm0.09}$ | $1.73_{\pm0.04}$ | $2.01_{\pm0.06}$ | $0.72_{\pm0.04}$ |
| CL4SRec | $4.45_{\pm0.04}$ | $1.86_{\pm0.02}$ | $4.25_{\pm0.10}$ | $2.01_{\pm0.09}$ | $2.03_{\pm0.06}$ | $0.74_{\pm0.03}$ |
| IPS | $3.81_{\pm0.05}$ | $1.73_{\pm0.03}$ | $3.73_{\pm0.03}$ | $1.40_{\pm0.05}$ | $1.66_{\pm0.04}$ | $0.64_{\pm0.02}$ |
| AdaRanker | $3.74_{\pm0.06}$ | $1.67_{\pm0.04}$ | $4.14_{\pm0.09}$ | $1.89_{\pm0.05}$ | $1.70_{\pm0.04}$ | $0.61_{\pm0.02}$ |
| STEAM | $4.69_{\pm0.06}$ | $1.76_{\pm0.02}$ | $4.98_{\pm0.05}$ | $2.90_{\pm0.02}$ | $1.75_{\pm0.02}$ | $0.69_{\pm0.02}$ |
| SSDRec | $4.52_{\pm0.07}$ | $1.95_{\pm0.03}$ | $4.19_{\pm0.08}$ | $3.28_{\pm0.06}$ | $2.03_{\pm0.06}$ | $0.72_{\pm0.03}$ |
| DiffuASR | $4.48_{\pm0.03}$ | $1.92_{\pm0.02}$ | $4.53_{\pm0.02}$ | $3.30_{\pm0.03}$ | $2.05_{\pm0.02}$ | $0.71_{\pm0.02}$ |
| CaDiRec | $5.05_{\pm0.05}$ | $2.21_{\pm0.10}$ | $2.56_{\pm0.04}$ | $1.79_{\pm0.03}$ | $2.14_{\pm0.05}$ | $0.72_{\pm0.07}$ |
| PDRec | $\underline{6.22}_{\pm0.03}$ | $\underline{3.17}_{\pm0.02}$ | $4.42_{\pm0.03}$ | $3.55_{\pm0.04}$ | $2.10_{\pm0.03}$ | $0.74_{\pm0.02}$ |
| DiffRec | $4.33_{\pm0.02}$ | $1.84_{\pm0.01}$ | $3.74_{\pm0.08}$ | $1.77_{\pm0.05}$ | $1.82_{\pm0.03}$ | $0.65_{\pm0.09}$ |
| DiffRIS | $4.51_{\pm0.03}$ | $1.95_{\pm0.02}$ | $4.28_{\pm0.03}$ | $2.03_{\pm0.04}$ | - | - |
| DreamRec | $4.78_{\pm0.06}$ | $2.23_{\pm0.02}$ | $\underline{5.16}_{\pm0.05}$ | $\underline{4.11}_{\pm0.02}$ | $\underline{2.26}_{\pm0.07}$ | $\underline{0.79}_{\pm0.01}$ |
| DreamMiss | $\mathbf{6.90}_{\pm0.01}$ | $\mathbf{4.34}_{\pm0.03}$ | $\mathbf{5.48}_{\pm0.02}$ | $\mathbf{4.77}_{\pm0.04}$ | $\mathbf{2.65}_{\pm0.03}$ | $\mathbf{0.88}_{\pm0.04}$ |
| improv. | 9.85% | 26.95% | 5.84% | 13.84% | 14.72% | 10.23% |

## 4.3 Ablation Study (RQ2)

The two probability models employed in DreamMiss are expected to capture the dynamic preference evolution, preserving user preference when simulating data missing with dual-side Thompson sampling. To evaluate their impact, we conduct ablation studies with eight variations of DreamMiss. The experimental results are shown in Table 2, where "w/o L" and "w/o G" indicate variants where the local or global probability model is replaced with random sampling, "w/o GL" denotes replacing both models with random sampling, "Base" represents generating oracle items without simulating data missing. To extend our method, we propose other metrics for probability models and compare their performance. Specifically, "popularity" is measured by the frequency of each item in all interactions, which reflects popularity bias. "Diversity" within a sequence is calculated to indicate exposure bias. "Seq-len" (Sequence length) serves as a measure of user activity, while "Item-pos" (item position) reflects short-term and long-term preferences. All of these metrics can serve as probability measures for a specific type of missing data.

As can be seen, all variations incorporating data missing simulation (*i.e.,* "w/o GL", "w/o G", "w/o L", "popularity", "Diversity", "Seq-len", and "Item-pos") outperform the "Base" model, highlighting the effectiveness of simulating data missing in enhancing the robustness of DDMs. Furthermore, both variations "w/o L" and "w/o G" exhibit superior performance compared to "w/o GL", while DreamMiss outperforms all variations. This demonstrates that sampling based on probability models with item continuity and sequence stability metrics can effectively maintain user preferences, thereby improving DDMs' recommendation performance through the strategic simulation of missing data. Additionally, DreamMiss (with item continuity and sequence stability metrics) achieves superior performance to other variations (with "popularity", "Diversity", "Seq-len", and "Item-pos" metrics), demonstrating the reliability of continuity and stability metrics for probability models. The reliability and robustness of the continuity and stability metrics lie in that they can create missing data regardless of the specific factors (*e.g.,* popularity bias or diversity bias), as long as such missing data does not lead to changes in user preferences.

Table 2: Ablation Study for the metrics of probability models. The best performance is bolded. H@20 and N@20 represent HR@20 and NDCG@20 respectively.

| Methods | YooChoose | | KuaiRec | | Zhihu | |
|---|---|---|---|---|---|---|
| | H@20(%) | N@20(%) | H@20(%) | N@20(%) | H@20(%) | N@20(%) |
| Base | $4.78_{\pm 0.06}$ | $2.23_{\pm 0.02}$ | $5.16_{\pm 0.05}$ | $4.11_{\pm 0.02}$ | $2.26_{\pm 0.07}$ | $0.79_{\pm 0.01}$ |
| w/o GL | $6.24_{\pm 0.07}$ | $3.91_{\pm 0.06}$ | $5.37_{\pm 0.05}$ | $4.19_{\pm 0.06}$ | $2.30_{\pm 0.05}$ | $0.80_{\pm 0.02}$ |
| w/o L | $6.41_{\pm 0.06}$ | $4.26_{\pm 0.05}$ | $5.44_{\pm 0.03}$ | $4.63_{\pm 0.02}$ | $2.34_{\pm 0.03}$ | $0.86_{\pm 0.02}$ |
| w/o G | $6.48_{\pm 0.01}$ | $4.29_{\pm 0.04}$ | $5.43_{\pm 0.04}$ | $4.64_{\pm 0.02}$ | $2.44_{\pm 0.02}$ | $0.81_{\pm 0.08}$ |
| popularity | $6.28_{\pm 0.02}$ | $4.18_{\pm 0.03}$ | $5.46_{\pm 0.05}$ | $4.57_{\pm 0.08}$ | $2.38_{\pm 0.03}$ | $0.80_{\pm 0.07}$ |
| Diversity | $6.26_{\pm 0.04}$ | $4.20_{\pm 0.06}$ | $5.45_{\pm 0.02}$ | $4.52_{\pm 0.02}$ | $2.29_{\pm 0.02}$ | $0.81_{\pm 0.03}$ |
| Seq-len | $6.27_{\pm 0.03}$ | $4.30_{\pm 0.06}$ | $5.46_{\pm 0.03}$ | $4.54_{\pm 0.08}$ | $2.30_{\pm 0.08}$ | $0.83_{\pm 0.06}$ |
| Item-pos | $6.28_{\pm 0.04}$ | $3.96_{\pm 0.02}$ | $5.20_{\pm 0.03}$ | $4.55_{\pm 0.04}$ | $2.24_{\pm 0.08}$ | $0.79_{\pm 0.05}$ |
| DreamMiss | $\mathbf{6.90}_{\pm 0.01}$ | $\mathbf{4.34}_{\pm 0.03}$ | $\mathbf{5.48}_{\pm 0.02}$ | $\mathbf{4.77}_{\pm 0.04}$ | $\mathbf{2.65}_{\pm 0.03}$ | $\mathbf{0.88}_{\pm 0.04}$ |

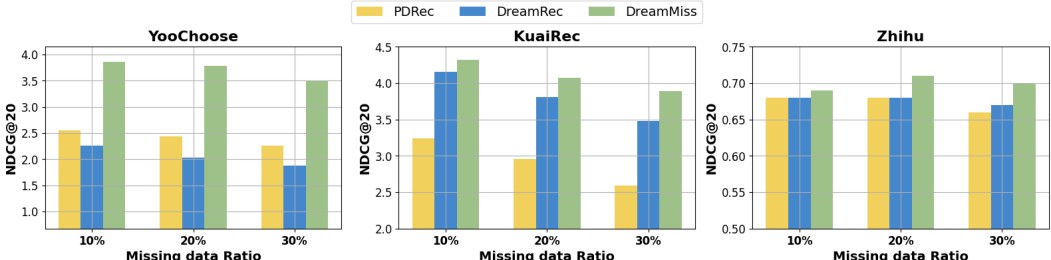

Figure 3: Performance of DreamMiss on synthetic datasets with different missing ratios.

## 4.4 ROBUSTNESS OF DREAMMISS (RQ3)

To validate the robustness of DreamMiss to various missing data ratios, we create synthetic datasets with missing data proportions of $10\%$, $20\%$, and $30\%$ respectively. We compare the recommendation performance of DreamMiss with representative baselines, including PDRec and DreamRec, across synthetic datasets. PDRec is an imputation-based algorithm that utilizes DDMs to enhance traditional recommenders, while DreamRec is a generative recommender where DDMs generate items to recommend directly. As shown in Figure 3, DreamMiss outperforms these baseline models on our synthetic datasets with varying missing ratios. Furthermore, the performance decline in DreamMiss due to increased missing data is less significant than that observed in the baseline models, as evidenced by the increasing height difference between the columns. These results demonstrate the superior robustness of DreamMiss to varying degrees of missing data, as well as the effectiveness of dual-side Thompson sampling in enhancing the robustness of DDMs.

## 4.5 SENSITIVITY ANALYSIS (RQ4)

Given that the guidance signal is crucial for steering the reverse process in DreamMiss towards personalized generation, we here investigate the model's sensitivity to the guidance strength parameter $w$. As illustrated in Figure 4, increasing the value of $w$ initially improves recommendation accuracy. This validates that utilizing guidance can enhance personalization. However, further increases in $w$ beyond an optimal point lead to a decline in performance, suggesting that overly strong guidance may compromise the quality of the generated oracle items. Besides, DreamMiss consistently outperforms the variation "random" which simulates missing data without considering preference evolution, further underscoring the importance of our dual-side probability models. Additionally, we examine the sensitivity of DreamMiss to the parameters of threshold $\lambda_1$ and $\lambda_2$, which determine the proportion of removed items when simulating data missing. The experimental results and analysis are presented in Appendix F.2 due to space constraints.

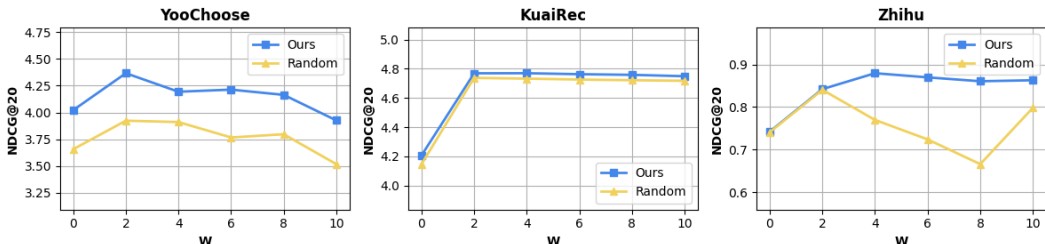

Figure 4: Sensitivity of DreamMiss to the strength of classifier-free guidance on multiple datasets, where "random" represents the variation "w/o GL" of DreamMiss.

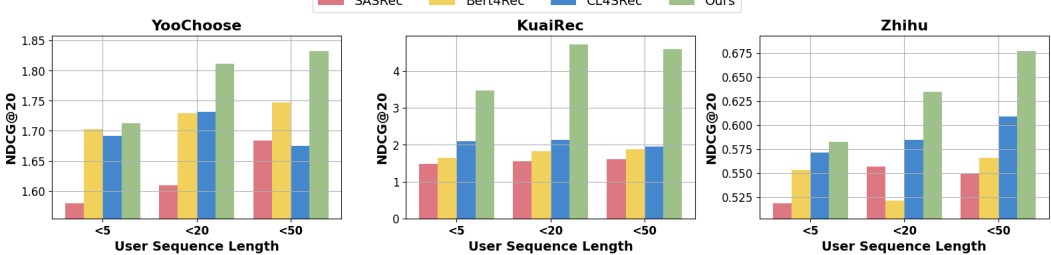

Figure 5: Performance of DreamMiss on multiple datasets with different sequence lengths.

### 4.6 GENERALIZATION ABILITY OF DREAMMISS (RQ5)

We further investigate the generalization ability of DreamMiss on datasets with different sequence lengths (*i.e.,* <5, <20, <50). We compare the recommendation performance of DreamMiss with representative baseline models, including SASRec, Bert4Rec, and CL4SRec. The results are shown in Figure 5. DreamMiss maintains a significant lead on different sequence lengths, validating the generalization ability of DreamMiss to different sequence lengths.

## 5 LIMITAIONS AND FUTURE WORK

The dynamics of user preferences may exhibit intricate patterns that transcend the capabilities of our dual-side probability models. Future research could benefit from integrating a more sophisticated understanding of user preference evolution into these models. Additionally, the issue of missing data, which can stem from various factors, such as exposure bias or popularity bias, presents an opportunity for targeted simulation to enhance the debiasing capabilities of DDMs.

## 6 CONCLUSION

In this paper, we propose DreamMiss, a novel approach that enhances the robustness of denoising diffusion models (DDMs) to uncertain missing data in sequential recommendation. By introducing a dual-side Thompson sampling strategy with local and global probability models, DreamMiss effectively simulates missing data while preserving user preference evolution, creating uncertainty-aware guidance for DDMs. We explain that incorporating such uncertainty-aware guidance for DDMs can promote consistency regularization, thus resiling against the uncertain missing data. Extensive experiments demonstrate that DreamMiss significantly outperforms multiple baselines, showcasing its potential to improve recommendation performance in real-world scenarios.

### ETHICS STATEMENT.

This work is designed for the generative sequential recommendation with DDMs, simulating missing data with a dual-side Thompson sampling strategy to create uncertainty-aware guidance. It focuses on mitigating the influence of missing data and enhancing DDMs' robustness. We do not foresee any direct, immediate, or negative societal impacts stemming from the outcomes of our research.

REPRODUCIBILITY STATEMENT.

All the results in this work are reproducible. We have discussed the hyperparameters search space and the details on devices and software environments in Appendix E.2. The implementation code is available at https://anonymous.4open.science/r/DreamMiss-E634, along with the hyperparameters we employed.

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

# A  RELATED WORK

In this section, we provide a review of missing data in recommendation and generative recommenders.

**Missing Data in Recommendation** can be classified into Missing At Random (MAR) without bias and Missing Not at Random (MNAR) with bias (Zheng et al., 2022; Wang et al., 2023b). In addressing missing data in sequential recommendation systems, existing methods can be broadly categorized into two types: latent variable decomposition (Saito et al., 2020; Li et al., 2023b), which aims to debias the data, and imputation techniques that complete the sequence with missing data (Steck, 2013; Hikmawati et al., 2024; Wang et al., 2019; Steck, 2010). For instance, Inverse Propensity Score (IPS) approaches (Schnabel et al., 2016; Li et al., 2023a; Wang et al., 2022) assign weights based on propensity scores to mitigate the bias. PDRec (Ma et al., 2024) leverages the DDMs to generate supplement items to the observed sequences. Here we emphasize the uncertain nature (Wang et al., 2023a; Fan et al., 2022b) of missingness (no matter MAR or MNAR) within the observed sequences by simulating it stochastically.

**Generative Recommenders** offer distinct advantages over discriminative recommenders, particularly in sequential recommendation tasks. Since GANs and VAEs are limited in the stability and quality of generation (Wang et al., 2023c; Zhao et al., 2024), diffusion models (DDMs) have emerged as a promising alternative, excelling at modeling complex data distributions and generating oracle items (Yang et al., 2023; Niu et al., 2024). For example, DiffuRec (Li et al., 2024) and DreamRec (Yang et al., 2023) generate the next items directly by corrupting them with noise and denoising based on the historical sequence. Additionally, DiffuASR (Liu et al., 2023a) and CaDiRec (Cui et al., 2024) address data sparsity by generating sequences or items with DDMs. In our work, we emphasize the quality of DDMs' guidance, tackling challenges posed by uncertain missing data and enhancing the robustness through consistency regularization.

# B  MOTIVATION AND EXPLANATION FOR OUR METHOD

## B.1  MOTIVATION FOR SIMULATING UNCERTAIN MISSING DATA

We denote the original missing data as $\delta$, the observed sequence as $\mathbf{e}_{1:N-1}$, the simulated missing data as $\delta'$, the complete sequence $\mathbf{e}_{1:N-1} \oplus \delta$, and the simulated sequence as $\mathbf{e}_{1:N-1} \ominus \delta'$.

- **Unavailability of complete sequence:** Due to the lack of prior knowledge for the complete sequence $\mathbf{e}_{1:N-1} \oplus \delta$ from an omniscient perspective, we cannot pinpoint where the missing data $\delta$ occurs or what specific content it pertains to.

- **Uncertainty of missing data:** Due to the variability of user preferences, the missing data $\delta$ is uncertain in the observed interaction sequence $\mathbf{e}_{1:N-1}$. User interaction may miss anywhere stochastically in the observed sequence.

Thus, compensating for missing data may introduce additional errors as shown in Figure 1 (a). The unavailability of the complete sequence $\mathbf{e}_{1:N-1} \oplus \delta$ and the uncertainty of missing data motivates us to simulate additional missing data $\delta'$ stochastically from the observed sequence $\mathbf{e}_{1:N-1}$, treating $\mathbf{e}_{1:N-1}$ as the complete sequence for the simulated sequence $\mathbf{e}_{1:N-1} \ominus \delta'$.

## B.2  MOTIVATION FOR THOMPSON SAMPLING

To simulate missing data stochastically while maintaining user preference, we propose Dual-side Thompson Sampling. The motivation for DTS is as follows:

- **Characteristics of TS:** TS strategy can balance the exploration and exploitation based on the known user preference in probability models. To capture user preference evolution, the probability models are established based on item continuity and sequence stability metrics. The Dual-side Thompson Sampling is more likely to remove items with high continuity in sequence with greater stability, according to the probability models. Such removal has little impact on understanding user preference evolution, as evidenced by the consistent green curves between the observed sequence and the one edited by DTS in Figure 1.

Figure 6: The comparison of different sampling strategies for simulating uncertain missing data.

- **Comparison with other sampling strategies:** We compare DTS with different sampling strategies, such as random-based sampling, time-based sampling, and model-based sampling. Random sampling samples items uniformly without exploiting known information, making it difficult to capture key patterns. Time-based sampling samples items based on the timestamps of interaction, which may overlook long-term or short-term preference. Model-based sampling samples items based on another trained model, which requires additional training resources and lacks supervision signals. All three strategies may disrupt users' original preference evolution when sampling, as the green curves of user preference evolution change in (a), (b), and (c) of Figure B.2.

### B.3 THEORETICAL ANALYSIS FOR OUR METHOD IN ADDRESSING MISSING DATA

Here we justify our method in simulating missing data to address missing data issues, which can realize diffusion models' consistency regularization and enhance the robustness through extrapolation. **Extrapolation:** Let $\hat{\mathbf{g}}$, $\bar{\mathbf{g}}$, and $\tilde{\mathbf{g}}$ denote the guidance encoded from observed sequence $\mathbf{e}_{1:N-1}$, unavailable complete sequence $\mathbf{e}_{1:N-1} \oplus \delta$ and simulated sequence $\mathbf{e}_{1:N-1} \ominus \delta'$, respectively. Our objective is to demonstrate the validity of the extrapolation, specifically the inequality $\|f_\theta(\mathbf{e}_N^{\tau_s}, \bar{\mathbf{g}}, \tau_s) - f_\theta(\mathbf{e}_N^{\tau_s}, \hat{\mathbf{g}}, \tau_s)\| \leq C\|f_\theta(\mathbf{e}_N^{\tau_s}, \hat{\mathbf{g}}, \tau_s) - f_\theta(\mathbf{e}_N^{\tau_s}, \tilde{\mathbf{g}}, \tau_s)\|$ for some constant $C$, where $\mathbf{e}_N^{\tau_s}$ represents the next interacted item with noise of $\tau_s$ time steps, $f_\theta$ is the denoising model.

Applying Taylor's Formula, we can express the two functions as follows:

$$f_\theta(\mathbf{e}_N^{\tau_s}, \bar{\mathbf{g}}, \tau_s) = f_\theta(\mathbf{e}_N^{\tau_s}, \hat{\mathbf{g}}, \tau_s) + (\bar{\mathbf{g}} - \hat{\mathbf{g}})^\top \nabla_g f_\theta(\mathbf{e}_N^{\tau_s}, \hat{\mathbf{g}}, \tau_s) + o(\|\bar{\mathbf{g}} - \hat{\mathbf{g}}\|), \tag{18}$$

$$f_\theta(\mathbf{e}_N^{\tau_s}, \tilde{\mathbf{g}}, \tau_s) = f_\theta(\mathbf{e}_N^{\tau_s}, \hat{\mathbf{g}}, \tau_s) + (\tilde{\mathbf{g}} - \hat{\mathbf{g}})^\top \nabla_g f_\theta(\mathbf{e}_N^{\tau_s}, \hat{\mathbf{g}}, \tau_s) + o(\|\tilde{\mathbf{g}} - \hat{\mathbf{g}}\|). \tag{19}$$

Then, combining the above two equalities, we have

$$f_\theta(\mathbf{e}_N^{\tau_s}, \bar{\mathbf{g}}, \tau_s) - f_\theta(\mathbf{e}_N^{\tau_s}, \hat{\mathbf{g}}, \tau_s) \tag{20}$$

$$= (\bar{\mathbf{g}} - \hat{\mathbf{g}})^\top \nabla f_\theta(\mathbf{e}_N^{\tau_s}, \hat{\mathbf{g}}, \tau_s) + o(\|\bar{\mathbf{g}} - \hat{\mathbf{g}}\|) \tag{21}$$

$$= \frac{(\bar{\mathbf{g}} - \hat{\mathbf{g}})^\top \nabla_g f_\theta(\mathbf{e}_N^{\tau_s}, \hat{\mathbf{g}}, \tau_s)}{(\tilde{\mathbf{g}} - \hat{\mathbf{g}})^\top \nabla_g f_\theta(\mathbf{e}_N^{\tau_s}, \hat{\mathbf{g}}, \tau_s)} (\tilde{\mathbf{g}} - \hat{\mathbf{g}})^\top \nabla_g f_\theta(\mathbf{e}_N^{\tau_s}, \hat{\mathbf{g}}, \tau_s) + o(\|\bar{\mathbf{g}} - \hat{\mathbf{g}}\|) \tag{22}$$

$$= \frac{(\bar{\mathbf{g}} - \hat{\mathbf{g}})^\top \nabla_g f_\theta(\mathbf{e}_N^{\tau_s}, \hat{\mathbf{g}}, \tau_s)}{(\tilde{\mathbf{g}} - \hat{\mathbf{g}})^\top \nabla_g f_\theta(\mathbf{e}_N^{\tau_s}, \hat{\mathbf{g}}, \tau_s)} (f_\theta(\mathbf{e}_N^{\tau_s}, \tilde{\mathbf{g}}, \tau_s) - f_\theta(\mathbf{e}_N^{\tau_s}, \hat{\mathbf{g}}, \tau_s)) + o(\|\bar{\mathbf{g}} - \hat{\mathbf{g}}\|) + o(\|\tilde{\mathbf{g}} - \hat{\mathbf{g}}\|), \tag{23}$$

where we assume that $(\tilde{\mathbf{g}} - \hat{\mathbf{g}})^\top \nabla_g f_\theta(\mathbf{e}_N^{\tau_s}, \hat{\mathbf{g}}, \tau_s) \neq 0$. Thus, we obtain the following inequality:

$$\|f_\theta(\mathbf{e}_N^{\tau_s}, \bar{\mathbf{g}}, \tau_s) - f_\theta(\mathbf{e}_N^{\tau_s}, \hat{\mathbf{g}}, \tau_s)\| \tag{24}$$

$$\leq \left| \frac{(\bar{\mathbf{g}} - \hat{\mathbf{g}})^\top \nabla_g f_\theta(\mathbf{e}_N^{\tau_s}, \hat{\mathbf{g}}, \tau_s)}{(\tilde{\mathbf{g}} - \hat{\mathbf{g}})^\top \nabla_g f_\theta(\mathbf{e}_N^{\tau_s}, \hat{\mathbf{g}}, \tau_s)} \right| \|f_\theta(\mathbf{e}_N^{\tau_s}, \hat{\mathbf{g}}, \tau_s) - f_\theta(\mathbf{e}_N^{\tau_s}, \tilde{\mathbf{g}}, \tau_s)\| + o(\|\bar{\mathbf{g}} - \hat{\mathbf{g}}\|) + o(\|\tilde{\mathbf{g}} - \hat{\mathbf{g}}\|),$$

$$\tag{25}$$

where $\|f_\theta(\mathbf{e}_N^{\tau_s}, \hat{\mathbf{g}}, \tau_s) - f_\theta(\mathbf{e}_N^{\tau_s}, \tilde{\mathbf{g}}, \tau_s)\|$ is the distance between the prediction from observed sequence and simulated sequence. To bound the coefficients $\left| \frac{(\bar{\mathbf{g}} - \hat{\mathbf{g}})^\top \nabla_g f_\theta(\mathbf{e}_N^{\tau_s}, \hat{\mathbf{g}}, \tau_s)}{(\tilde{\mathbf{g}} - \hat{\mathbf{g}})^\top \nabla_g f_\theta(\mathbf{e}_N^{\tau_s}, \hat{\mathbf{g}}, \tau_s)} \right|$, we need to analyze the two inner products, $(\bar{\mathbf{g}} - \hat{\mathbf{g}})^\top \nabla_g f_\theta(\mathbf{e}_N^{\tau_s}, \hat{\mathbf{g}}, \tau_s)$ and $(\tilde{\mathbf{g}} - \hat{\mathbf{g}})^\top \nabla_g f_\theta(\mathbf{e}_N^{\tau_s}, \hat{\mathbf{g}}, \tau_s)$. If we can simulate the mechanism of missing data—specifically, if the missing data process from $\mathbf{e}_{1:N-1}$ to $\mathbf{e}_{1:N-1} \ominus \delta'$ can align well with that from $\mathbf{e}_{1:N-1} \oplus \delta$ to $\mathbf{e}_{1:N-1}$—the difference between the two pair of data will be roughly equivalent. Consequently, the two differences in guidance, $\bar{\mathbf{g}} - \hat{\mathbf{g}}$ and $\hat{\mathbf{g}} - \tilde{\mathbf{g}}$, will also

Table 3: The rationale of DDPM.

| | |
|---|---|
| Forward Process | $q(x_t|x_0) = \mathcal{N}(x_t; \sqrt{\bar{\alpha}_t}x_0, (1-\bar{\alpha}_t)I)$ |
| Reverse Process | $q(\mathbf{x}^{t-1}|\mathbf{x}^t, \mathbf{x}^0) = \mathcal{N}(\mathbf{x}^{t-1}; \tilde{\boldsymbol{\mu}}_t(\mathbf{x}^t, \mathbf{x}^0), \tilde{\beta}_t\mathbf{I})$ |
| | $\tilde{\boldsymbol{\mu}}_t(\mathbf{x}^t, \mathbf{x}^0) = \frac{\sqrt{\bar{\alpha}_{t-1}}\beta_t}{1-\bar{\alpha}_t}\mathbf{x}^0 + \frac{\sqrt{\bar{\alpha}_t}(1-\bar{\alpha}_t)}{1-\bar{\alpha}_t}\mathbf{x}^t, \tilde{\beta}_t = \frac{1-\bar{\alpha}_{t-1}}{1-\bar{\alpha}_t}\beta_t$ |
| Training Objective | $L_\gamma(\epsilon_\theta) := \sum_{t=1}^{T} \gamma_t \mathbb{E}_{\boldsymbol{x}^0 \sim q(\boldsymbol{x}^0), \epsilon_t \sim \mathcal{N}(\mathbf{0},\boldsymbol{I})} \left[ \left\| \epsilon_\theta^{(t)}(\sqrt{\alpha_t}\boldsymbol{x}^0 + \sqrt{1-\alpha_t}\epsilon_t) - \epsilon_t \right\|_2^2 \right]$ |
| Sampling Steps | $\{1, ..., T\}$ |

Table 4: The rationale of DDIM.

| | |
|---|---|
| Forward Process | $q(x_t|x_0) = \mathcal{N}(x_t; \sqrt{\bar{\alpha}_t}x_0, (1-\bar{\alpha}_t)I)$ |
| Reverse Process | $q_\sigma(\boldsymbol{x}^{t-1}|\boldsymbol{x}^t, \boldsymbol{x}^0) = \mathcal{N}\left(\sqrt{\alpha_{t-1}}\boldsymbol{x}^0 + \sqrt{1-\alpha_{t-1}-\sigma_t^2} \cdot \frac{\boldsymbol{x}^t - \sqrt{\alpha_t}\boldsymbol{x}^0}{\sqrt{1-\alpha_t}}, \sigma_t^2\boldsymbol{I}\right)$ |
| | $\sigma_t = \eta\sqrt{\frac{1-\alpha_{t-1}}{1-\alpha_t}}\sqrt{1-\frac{\alpha_t}{\alpha_{t-1}}}$ |
| Training Objective | $L_\gamma(\epsilon_\theta) := \sum_{t=1}^{T} \gamma_t \mathbb{E}_{\boldsymbol{x}^0 \sim q(\boldsymbol{x}^0), \epsilon_t \sim \mathcal{N}(\mathbf{0},\boldsymbol{I})} \left[ \left\| \epsilon_\theta^{(t)}(\sqrt{\alpha_t}\boldsymbol{x}^0 + \sqrt{1-\alpha_t}\epsilon_t) - \epsilon_t \right\|_2^2 \right]$ |
| Sampling Steps | $\{\tau_1, ..., \tau_S\}$ which is a sub-sequence of $\{1, ..., T\}$ |

be approximately equal. In this scenario, the coefficient will be close to 1, resulting in a bounded value $C > 0$.

**Consistency regularization:** To ensure the effectiveness of extrapolation, we leverage consistency regularization to minimize $\|f_\theta(\mathbf{e}_N^{\tau_s}, \hat{\mathbf{g}}, \tau_s) - f_\theta(\mathbf{e}_N^{\tau_s}, \tilde{\mathbf{g}}, \tau_s)\|$. Since the interaction sequences are edited stochastically across different epochs, $\hat{\mathbf{g}}$ and $\tilde{\mathbf{g}}$ can serve as the perturbated pairs. Let the ground-truth label of the next item be $\mathbf{e}_N^0$. By completing the square, we obtain the inequality:

$$\|f_\theta(\mathbf{e}_N^{\tau_s}, \hat{\mathbf{g}}, \tau_s) - f_\theta(\mathbf{e}_N^{\tau_s}, \tilde{\mathbf{g}}, \tau_s)\|_2^2 \leq 2\left(\|f_\theta(\mathbf{e}_N^{\tau_s}, \hat{\mathbf{g}}, \tau_s)\mathbf{e}_N^0\|_2^2 + \|f_\theta(\mathbf{e}_N^{\tau_s}, \tilde{\mathbf{g}}, \tau_s) - \mathbf{e}_N^0\|_2^2\right). \quad (26)$$

Consequently, we can achieve consistency regularization of minimizing the left-hand side by minimizing the right-hand side, which stems from our reconstruction loss.

## C  RATIONALE FOR DDPM AND DDIM

Here we explain the different rationale of DDPM (Ho et al., 2020) and DDIM (Song et al., 2021). For DDPM, the forward process and reverse processes are both Markovian chains. The number of iterative steps in the reverse process is equal to that in the forward process due to the reliance on the Markov assumption. To ensure the generation quality, DDPM needs a substantial number of sampling steps in the reverse process. Unlike DDPM, DDIM accelerates the reverse process by relaxing the Markov assumption, achieving comparable generation quality with significantly fewer sampling steps. Below we list the rationale of DDPM and DDIM in Table 3 and Table 4, respectively.

Both DDPM and DDIM are trained under the same objective, the difference lies in the reverse process of the inference phase. For DDIM, the $\sigma_t = \eta\sqrt{\frac{1-\bar{\alpha}_{t-1}}{1-\bar{\alpha}_t}}\sqrt{1-\frac{\bar{\alpha}_t}{\bar{\alpha}_{t-1}}}$ is an arbitrary vector. When $\eta = 0$, the sampling process becomes a deterministic process as DDIM, and when $\eta = 1$, it corresponds to DDPM. Since the training objective does not depend on the specific forward process as long as $q_\sigma(x_t|x_0)$ is fixed. The deterministic sampling process of DDIM does not rely on the Markov chain, so it can adopt a forward process with steps fewer than $T$. Thus the step-by-step sampling process in DDPM, which operates over $1 \sim T$, can be modified to iterate over a sub-sequence $\{\tau_1, ..., \tau_S\}$ of $\{1, ..., T\}$. This change enables the acceleration of the sampling process of DDIM.

## D  ALGORITHM

Here we list the algorithm of DreamMiss's training phase in the batch $\mathcal{B}_{tr}$.

---

**Algorithm 1:** Training phase of DreamMiss

---

**Input:** $\mathbf{e}_{1:N-1}, \mathbf{e}_N$ from $\mathcal{B}_{tr}$, hyperparameters $w, \rho, \lambda_1, \lambda_2$, variance schedule $[\alpha_{\tau_s}]_{s=1}^S$
**Output:** updated denoise model $f_\theta(\cdot, \cdot, \cdot)$.
1: $\text{con}_n \leftarrow$ Equation 6, $\text{sta}_k \leftarrow$ Equation 8       ▷ Obtain continuity and stability scores.
2: $\hat{p}_n \sim L(\text{con}_n, p_n), \quad \hat{p}_k \sim G(\text{sta}_k, p_k)$       ▷ Sampling based on the two probability models.
3: $\mathbf{e}'_{1:N-1} \leftarrow$ Equation 10       ▷ Edit the sequence with $\lambda_1, \lambda_2$.
4: $\mathbf{g} = \textbf{T-enc}(\mathbf{e}'_{1:N-1})$ or $\Phi$ with probability $\rho$       ▷ Encode classifier-free guidance.
5: $\tau_s \sim [\tau_1, \ldots, \tau_S]$       ▷ Sample diffusion step.
6: $\boldsymbol{\epsilon} \sim \mathcal{N}(0, I)$       ▷ Sample Gaussian noise.
7: $\mathbf{e}_N^{\tau_s} = \sqrt{\alpha_{\tau_s}}\mathbf{e}_N^0 + \sqrt{1 - \alpha_{\tau_s}}\boldsymbol{\epsilon}$       ▷ Add Gaussian noise, $\mathbf{e}_N^0 = \mathbf{e}_N$.
8: $\theta = \theta - \mu\nabla_\theta \|\mathbf{e}_N^0 - f_\theta(\mathbf{e}_N^{\tau_s}, \mathbf{g}, \tau_s)\|^2$       ▷ Update the parameters of $f_\theta(\cdot, \cdot, \cdot)$.

---

Here we list the algorithm of the generating phase of DreamMiss.

---

**Algorithm 2:** Generating phase of DreamMiss

---

**Input:** Interaction sequence $\mathbf{e}_{1:N-1}$, hyperparameters $w$, optimal denoise model $f_\theta(\cdot, \cdot, \cdot)$.
**Output:** Oracle item embedding $\mathbf{e}_N^0$.
1: $\mathbf{e}_N^{\tau_S} \sim \mathcal{N}(0, I)$       ▷ Sample Gaussian noise.
2: $\mathbf{g} = \textbf{T-enc}(\mathbf{e}_{1:N-1})$       ▷ Encode observed sequence.
3: **for** $\tau_s = \tau_S, \ldots, \tau_1$ **do**
4:     $\tilde{f}_\theta(\mathbf{e}_N^{\tau_s}, \mathbf{g}, \tau_s) = (1 + w)f_\theta(\mathbf{e}_N^{\tau_s}, \mathbf{g}, \tau_s) - wf_\theta(\mathbf{e}_N^{\tau_s}, \Phi, \tau_s)$    ▷ Control the guidance strength.
5:     $\hat{\mathbf{e}}_N^0 = \tilde{f}_\theta(\mathbf{e}_N^{\tau_s}, \mathbf{g}, \tau_s)$
6:     $\mathbf{e}_N^{\tau_{s-1}} = \sqrt{\alpha_{\tau_{s-1}}}\hat{\mathbf{e}}_N^0 + \sqrt{1 - \alpha_{\tau_{s-1}}} \cdot \frac{\mathbf{e}_N^{\tau_s} - \sqrt{\alpha_{\tau_s}}\hat{\mathbf{e}}_N^0}{\sqrt{1-\alpha_{\tau_s}}}$       ▷ Reverse iteratively.
7: **end for**
8: **return** $\mathbf{e}_N^0$

---

# E DETAILS OF EXPERIMENTAL IMPLEMENTATION

## E.1 DETAILS OF DATASETS

The YooChoose dataset (Ben-Shimon et al., 2015) originates from the RecSys Challenge 2015. KuaiRec (Gao et al., 2022) is collected from the recommendation logs of a video-sharing mobile application, offering a rich source of user interaction records in a short video platform. Zhihu (Hao et al., 2021) is obtained from a social knowledge-sharing community, where users are presented with a list of posts and can click on their preferred content. After preprocessing, the statistics of the three datasets are listed in Table 5.

## E.2 DETAILS OF IMPLEMENTAION

Our experiments are implemented using Python 3.9 and PyTorch 2.0.1, with computations performed on Nvidia GeForce RTX 3090 GPUs. The model is optimized with AdamW. The dimension of item embeddings is 64 across all models. The learning rate is tuned within the range of $[0.01, 0.005, 0.001, 0.0005, 0.0001, 0.00005]$. For diffusion models, we varied the total diffusion step $T$ across $[500, 1000, 2000]$, employing intervals of 100 to obtain corresponding $\tau_S$ values of $[5, 10, 20]$ for DDIM. The guidance strength $w$ is set within the range $[0, 2, 4, 6, 8, 10]$, and the threshold $th_g, th_l$ are tuning across the range $[0, 0.1, \ldots, 1]$. We set the unconditional training probability $\rho$ as 0.1 suggested by Ni et al. (2022).

Table 5: Statistics of the three datasets.

| Dataset | YooChoose | KuaiRec | Zhihu |
|---|---|---|---|
| #sequences | 128,468 | 92,090 | 11,714 |
| #items | 9,514 | 7,261 | 4,838 |
| #interactions | 539,436 | 737,163 | 77,712 |

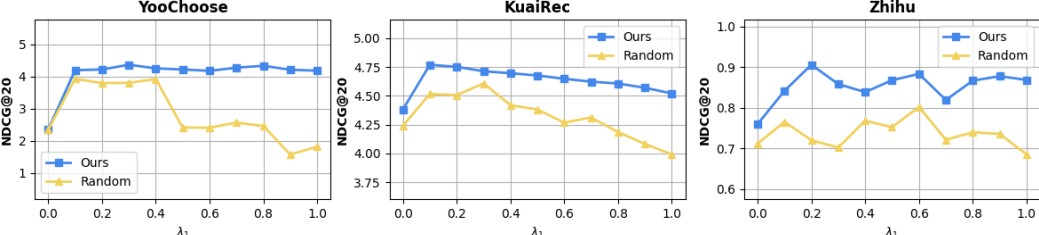

Figure 7: Sensitivity of DreamMiss to the hyperparameter of $\lambda_1$ on multiple datasets, demonstrating the proportion of edited sequences. The "random" represents the variation "w/o GL" of DreamMiss.

## F  EXPERIMENTAL RESULTS

### F.1  EXPERIMENTS OF DREAMMISS ON LARGER OR DIVERSE DATASETS.

To illustrate the effectiveness of DreamMiss on larger or diverse datasets, we conduct additional experiments on the Steam dataset, one of the largest datasets commonly used in research, as well as on the Amazon-Beauty and Amazon-Toys datasets, which come from different domains. We treat all ratings as implicit feedback and organize them chronologically by the timestamps. Items and users with fewer than 5 interactions are filtered. The max length of interaction history is set to 50. The statistics of this dataset are listed in Table 6. The baseline methods are the same as those in Table 1. As shown in Table 7, our method consistently outperforms various baselines on the larger dataset (Steam) and diverse datasets (Amazon-beauty and Amazon-toys), further highlighting the effectiveness of DreamMiss.

### F.2  SENSITIVITY ANALYSIS FOR THRESHOLD PARAMETERS

The parameters of threshold $\lambda_1$ and $\lambda_2$ represent the proportion of edited sequences and removed items when simulating data missing in DreamMiss. The experimental results that analyze DreamMiss's sensitivity to these two parameters are shown in Figure 7 and 8. The performance remains relatively stable; however, excessive or insufficient removal can result in suboptimal outcomes. This highlights the importance of choosing appropriate thresholds for dual-side Thompson sampling to create uncertainty-aware guidance.

### F.3  ROBUSTNESS OF DREAMMISS ON SYNTHETIC DATASETS WITH MISSING DATA

Following the setting of synthetic datasets in Section 4.4, we conduct experiments to compare the three methods (*i.e.,* simulating missing data with DreamMiss, leveraging observed data with DreamRec,

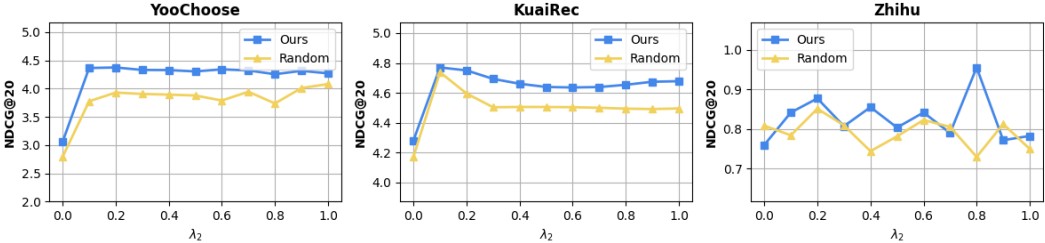

Figure 8: Sensitivity of DreamMiss to the hyperparameter of $\lambda_2$ on multiple datasets, demonstrating the proportion of removed items. The "random" represents the variation "w/o GL" of DreamMiss.

Table 6: Statistics of the additional datasets.

| Dataset | Steam | Beauty | Toys |
|---|---|---|---|
| #sequences | 281,428 | 22,363 | 19,4124 |
| #items | 13,044 | 12,101 | 11,924 |
| #interactions | 3,485,022 | 198,502 | 167,597 |

Table 7: Overall performance of different methods for the sequential recommendation on larger dataset (Steam) and diverse datasets from different domains (Beauty and Toys). The best score and the second-best score are bolded and underlined, respectively.

| Methods | Steam | | Toys | | Beauty | |
|---|---|---|---|---|---|---|
| | H@20(%) | N@20 (%) | H@20 (%) | N@20(%) | H@20(%) | N@20(%) |
| GRU4Rec | $9.23_{\pm 0.05}$ | $3.56_{\pm 0.03}$ | $3.18_{\pm 0.08}$ | $1.27_{\pm 0.03}$ | $3.85_{\pm 0.09}$ | $1.38_{\pm 0.06}$ |
| Caser | $15.20_{\pm 0.09}$ | $6.62_{\pm 0.05}$ | $8.83_{\pm 0.09}$ | $4.02_{\pm 0.05}$ | $8.67_{\pm 0.06}$ | $4.36_{\pm 0.10}$ |
| SASRec | $13.61_{\pm 0.06}$ | $5.36_{\pm 0.08}$ | $9.23_{\pm 0.07}$ | $4.33_{\pm 0.02}$ | $8.98_{\pm 0.12}$ | $3.66_{\pm 0.07}$ |
| Bert4Rec | $12.73_{\pm 0.08}$ | $5.20_{\pm 0.07}$ | $4.59_{\pm 0.08}$ | $1.90_{\pm 0.06}$ | $5.79_{\pm 0.11}$ | $2.35_{\pm 0.12}$ |
| CL4SRec | $15.06_{\pm 0.08}$ | $6.12_{\pm 0.06}$ | $9.09_{\pm 0.03}$ | $5.08_{\pm 0.03}$ | $10.18_{\pm 0.11}$ | $4.85_{\pm 0.12}$ |
| IPS | $15.65_{\pm 0.08}$ | $6.46_{\pm 0.02}$ | $9.29_{\pm 0.01}$ | $5.27_{\pm 0.04}$ | $10.15_{\pm 0.02}$ | $4.56_{\pm 0.07}$ |
| AdaRanker | $15.71_{\pm 0.07}$ | $6.68_{\pm 0.08}$ | $8.18_{\pm 0.02}$ | $4.33_{\pm 0.02}$ | $8.03_{\pm 0.08}$ | $3.80_{\pm 0.06}$ |
| DiffuASR | $15.74_{\pm 0.04}$ | $6.59_{\pm 0.06}$ | $\underline{9.39}_{\pm 0.04}$ | $5.19_{\pm 0.06}$ | $10.03_{\pm 0.06}$ | $5.16_{\pm 0.11}$ |
| CaDiRe | $15.65_{\pm 0.07}$ | $6.42_{\pm 0.12}$ | $9.33_{\pm 0.03}$ | $5.16_{\pm 0.11}$ | $9.85_{\pm 0.08}$ | $4.46_{\pm 0.04}$ |
| PDRec | $\underline{15.78}_{\pm 0.07}$ | $6.51_{\pm 0.08}$ | $9.08_{\pm 0.08}$ | $5.12_{\pm 0.06}$ | $10.24_{\pm 0.06}$ | $\underline{5.02}_{\pm 0.09}$ |
| DiffRec | $15.09_{\pm 0.04}$ | $\underline{6.89}_{\pm 0.03}$ | $9.18_{\pm 0.06}$ | $\underline{5.25}_{\pm 0.04}$ | $10.21_{\pm 0.04}$ | $5.14_{\pm 0.02}$ |
| DreamRec | $15.08_{\pm 0.08}$ | $6.39_{\pm 0.08}$ | $9.18_{\pm 0.08}$ | $5.22_{\pm 0.08}$ | $\underline{10.32}_{\pm 0.03}$ | $4.88_{\pm 0.07}$ |
| DreamMiss | $\mathbf{16.19}_{\pm 0.01}$ | $\mathbf{7.52}_{\pm 0.03}$ | $\mathbf{9.88}_{\pm 0.01}$ | $\mathbf{5.39}_{\pm 0.03}$ | $\mathbf{10.72}_{\pm 0.06}$ | $\mathbf{5.40}_{\pm 0.04}$ |
| improv. | 2.53% | 8.38% | 4.96% | 2.60% | 3.73% | 7.04% |

and recovering missing data with DreamImp), to validate the superiority of DreamMiss over data completion methods. DreamImp is our imputation-based algorithm designed for diffusion models. In the absence of prior knowledge about the complete sequence, we use the average of two adjacent items for imputation at random positions. The detailed results are shown in Table 8. Experimental results indicate that the method of simulating missing data (DreamMiss) outperforms approaches that rely solely on observed data (DreamRec) or attempt to recover missing data (DreamImp). This further demonstrates the superiority of simulating missing data over recovering them without prior knowledge of the complete sequence.

Furthermore, to validate the effectiveness of DreamMiss in addressing different kinds of missing data, we construct two synthetic datasets: the "R" is constructed with missing data that occurs randomly and may not fully reflect user preferences; the "P" is designed with missing data aligned with user preferences. We conduct experiments on these two synthetic datasets, and the results are presented in Table 9. We can observe that DreamMiss is more effective on dataset "P" with a more significant improvement than on dataset "R". This validates that DreamMiss can effectively address the missing data issue when the sequence accurately reflects users' complete preferences.

### F.4 BROADER APPLICABILITY OF DUAL-SIDE THOMPSON SAMPLING

Introducing missing data with DTS can be considered as introducing a form of noise within the sequence, which can enhance DDMs' denoising ability to missing data rather than Gaussian noise. However, DTS can be **a general algorithm to enhance recommendation systems' robustness** against uncertain missing data. To validate the performance of DTS when extended to traditional recommenders, We conduct experiments on other recommenders, including GRU4Rec, SASRec, CL4Rec, Caser, AdaRanker, and DiffuASR. The experimental results are presented in Table 10.

As shown in Table 10, applying DTS to traditional recommenders can yield nearly universal improvement in recommendation performance. This empirically proves the effectiveness and extensibility of DTS. Notably, when applied to DreamRec, a generative recommender using DDMs, it yields the highest performance gains on average. This observation highlights that DDMs provide a solid

Table 8: Performance comparison for the three methods (*i.e.,* simulating missing data, leveraging observed data, and recovering missing data.

| Methods | YooChoose | | KuaiRec | | Zhihu | |
|---|---|---|---|---|---|---|
| | H@20(%) | N@20(%) | H@20(%) | N@20(%) | H@20(%) | N@20(%) |
| DreamRec (10%) | 5.11 | 2.16 | 4.99 | 4.16 | 1.95 | 0.68 |
| DreamImp (10%) | 5.32 | 2.28 | 4.89 | 4.07 | 1.98 | 0.68 |
| DreamMiss (10%) | **6.49** | **3.98** | **5.04** | **4.32** | **2.05** | **0.69** |
| DreamRec (20%) | 4.87 | 2.03 | 4.59 | 3.81 | 1.89 | 0.68 |
| DreamImp (20%) | 4.91 | 2.55 | 4.65 | 3.76 | 1.91 | 0.70 |
| DreamMiss (20%) | **6.26** | **3.66** | **4.77** | **4.07** | **2.22** | **0.71** |
| DreamRec (30%) | 4.74 | 1.97 | 4.18 | 3.48 | 1.89 | 0.67 |
| DreamImp (30%) | 4.73 | 2.18 | 4.21 | 3.59 | 1.93 | 0.68 |
| DreamMiss (30%) | **5.92** | **3.29** | **4.49** | **3.69** | **2.12** | **0.70** |

Table 9: Performance of DreamMiss and DreamRec on different missing data types.

| Methods | YooChoose | | KuaiRec | | Zhihu | |
|---|---|---|---|---|---|---|
| | H@20(%) | N@20(%) | H@20(%) | N@20(%) | H@20(%) | N@20(%) |
| DreamRec (R) | 5.11 | 2.16 | 4.99 | 4.16 | 1.95 | 0.68 |
| DreamMiss (R) | 6.49 | 3.98 | 5.04 | 4.32 | 2.05 | 0.69 |
| improv. (R) | 21.26% | 45.73% | 0.09% | 3.70% | 4.88% | 1.45% |
| DreamRec (P) | 5.15 | 2.17 | 4.93 | 4.01 | 2.08 | 0.70 |
| DreamMiss (P) | 6.67 | 4.12 | 5.37 | 4.72 | 2.21 | 0.76 |
| improv. (P) | 22.79% | 47.33% | 8.19% | 15.04% | 5.88% | 7.89% |

foundation for DTS to achieve DDMs' consistency regularization from the experimental perspective, owing to their capability in modeling complex data distributions and denoising the missing data.

## F.5 Performance Comparison between DTS and Other Sampling Strategies

We conducted experiments to compare the performance of different sampling strategies. Experimental results are presented in Table 11. The "Base" represents the diffusion-based recommender without a sampling strategy. The random-based sampling sample items uniformly, time-based, and model-based sampling strategies are introduced in Section B.2 and Figure B.2.

Observing the results in Table 11, different sampling strategies have a noticeable impact on the diffusion-based recommender compared to the "Base" approach. However, DreamMiss demonstrates superior performance than other sampling strategies, confirming the advantages of DTS.

## F.6 Abalation Study for DDPM and DDIM

DDIM is designed to accelerate the reverse process while maintaining the comparable performance of DDPM. We conduct experiments to demonstrate the effect of DDIM and DDPM on the performance of diffusion-based recommenders. Specifically, we apply DDPM and DDIM for DreamRec and DreamMiss respectively to validate their impact on the performance of diffusion-based recommenders. The experimental results are presented in Table 12.

As shown in Table 12, the performance of DDPM and DDIM is comparable, as evidenced by the similar results for DreamRec (DDPM) and DreamRec (DDIM). This validates that the number of iterative steps in the reverse phase will not cause performance improvement or degradation. Furthermore, DreamMiss consistently outperforms DreamRec, demonstrating the effectiveness of our DTS in enhancing the robustness of diffusion models. Thus it is the DTS that results in performance improvement instead of the accelerated sampling with fewer iterative steps in DDIM.

Table 10: Comparison of DreamMiss and traditional recommenders on performance improvement from dual-side Thompson sampling strategy (DTS).

| Methods | YooChoose | | KuaiRec | | Zhihu | |
|---|---|---|---|---|---|---|
| | H@20(%) | N@20 (%) | H@20 (%) | N@20(%) | H@20(%) | N@20(%) |
| GRU4Rec | 3.89 | 1.62 | 3.32 | 1.23 | 1.78 | 0.67 |
| + DTS | 3.96 | 1.72 | 3.43 | 1.40 | 1.83 | 0.68 |
| improv. | 1.77% | 5.81% | 3.21% | 12.14% | 2.73% | 1.47% |
| SASRec | 3.68 | 1.63 | 3.92 | 1.53 | 1.62 | 0.60 |
| +DTS | 3.98 | 1.58 | 3.96 | 1.63 | 1.70 | 0.72 |
| improv. | 7.54% | $-3.07\%$ | 1.01% | 6.13% | 4.71% | **11.11%** |
| CL4SRec | 4.45 | 1.86 | 4.25 | 2.01 | 2.03 | 0.74 |
| +DTS | 4.63 | 1.88 | 4.57 | 2.25 | 2.16 | 0.78 |
| improv. | 3.89% | 1.06% | 7.00% | 10.67% | 6.02% | 5.13% |
| Caser | 4.06 | 1.88 | 2.88 | 1.07 | 1.57 | 0.59 |
| +DTS | 4.27 | 2.01 | 3.19 | 1.09 | 1.72 | 0.65 |
| improv. | 4.92% | 6.47% | **9.72%** | 1.83% | 8.72% | 9.23% |
| AdaRanker | 3.74 | 1.67 | 4.14 | 1.89 | 1.70 | 0.61 |
| DTS | 4.16 | 1.97 | 4.33 | 1.93 | 1.64 | 0.67 |
| improv. | 10.10% | 15.22% | 4.39% | 2.07% | $-3.53\%$ | 8.96% |
| DiffuASR | 4.48 | 1.92 | 4.53 | 3.30 | 2.05 | 0.71 |
| +DTS | 4.66 | 2.08 | 4.58 | 3.98 | 2.05 | 0.73 |
| improv. | 3.86% | 7.69% | 1.09% | **17.09%** | 0.00% | 2.82% |
| DreamRec | 4.78 | 2.23 | 5.16 | 4.11 | 2.26 | 0.79 |
| +DTS | 6.90 | 4.34 | 5.48 | 4.77 | 2.65 | 0.88 |
| improv. | **30.72%** | **48.62%** | 5.84% | 13.84% | **14.72%** | 10.23% |

Table 11: The performance comparison of different sampling strategies to create uncertainty-aware guided diffusion for the recommendation.

| Methods | YooChoose | | KuaiRec | | Zhihu | |
|---|---|---|---|---|---|---|
| | H@20(%) | N@20 (%) | H@20 (%) | N@20(%) | H@20(%) | N@20(%) |
| Base | $4.78_{\pm0.06}$ | $2.23_{\pm0.02}$ | $5.16_{\pm0.05}$ | $4.11_{\pm0.02}$ | $2.26_{\pm0.07}$ | $0.79_{\pm0.01}$ |
| Random-based | $6.24_{\pm0.07}$ | $3.91_{\pm0.06}$ | $5.37_{\pm0.05}$ | $4.19_{\pm0.06}$ | $2.30_{\pm0.05}$ | $0.80_{\pm0.02}$ |
| Time-based | $6.28_{\pm0.04}$ | $3.96_{\pm0.02}$ | $5.20_{\pm0.03}$ | $4.55_{\pm0.04}$ | $2.24_{\pm0.08}$ | $0.79_{\pm0.05}$ |
| Model-based | $6.49_{\pm0.06}$ | $3.62_{\pm0.06}$ | $5.18_{\pm0.06}$ | $4.59_{\pm0.06}$ | $2.36_{\pm0.02}$ | $0.80_{\pm0.02}$ |
| DreamMiss | $\mathbf{6.90}_{\pm0.01}$ | $\mathbf{4.34}_{\pm0.03}$ | $\mathbf{5.48}_{\pm0.02}$ | $\mathbf{4.77}_{\pm0.04}$ | $\mathbf{2.65}_{\pm0.03}$ | $\mathbf{0.88}_{\pm0.04}$ |

### F.7 COMPUTAIONAL RESOURCE REQUIREMENT

**Computational complexity:** DreamMiss employs a Transformer as the sequence encoder and a multi-layer perceptron (MLP) as the denoising model. The Transformer model consists of a Self-Attention Mechanism, a Feed-Forward Neural Network, and Layer Normalization. Let $L$ be the length of the input sequence, $d$ be the embedding dimension, and $N$ be the number of layers in the model. The complexity for computing self-attention at each layer is $O(L^2 \cdot d)$, and the complexity for the MLP network is $O(L \cdot d^2)$. For $N$ layers, the overall computational complexity becomes $O(N \cdot (L^2 \cdot d + L \cdot d^2))$ in the training phase. Let $K$ be the number of reverse steps, the computational complexity of DreamMiss can be $O(K \cdot N \cdot (L^2 \cdot d + L \cdot d^2))$ in the inference phase.

**Complexity comparison:** The computational complexity of DreamMiss for training each epoch is nearly similar to other diffusion-based recommenders and traditional recommenders that use the same sequence encoder. Furthermore, since we employ DDIM to accelerate sampling during the inference phase, the value of $K$ for DreamMiss is significantly lower than that of DreamRec. As

Table 12: Performance comparison of DDPM and DDIM for diffusion-based recommenders.

| Methods | YooChoose | | KuaiRec | | Zhihu | |
|---|---|---|---|---|---|---|
| | H@20(%) | N@20 (%) | H@20 (%) | N@20(%) | H@20(%) | N@20(%) |
| DreamRec (DDPM) | $4.78_{\pm0.06}$ | $2.23_{\pm0.02}$ | $5.16_{\pm0.05}$ | $4.11_{\pm0.02}$ | $2.26_{\pm0.07}$ | $0.79_{\pm0.01}$ |
| DreamRec (DDIM) | $4.85_{\pm0.06}$ | $2.26_{\pm0.07}$ | $4.93_{\pm0.03}$ | $4.01_{\pm0.07}$ | $2.25_{\pm0.02}$ | $0.82_{\pm0.02}$ |
| DreamMiss (DDPM) | $6.82_{\pm0.02}$ | $4.33_{\pm0.02}$ | $5.49_{\pm0.02}$ | $4.74_{\pm0.08}$ | $2.45_{\pm0.03}$ | $0.85_{\pm0.02}$ |
| DreamMiss (DDIM) | $6.90_{\pm0.01}$ | $4.34_{\pm0.03}$ | $5.48_{\pm0.02}$ | $4.77_{\pm0.04}$ | $2.65_{\pm0.03}$ | $0.88_{\pm0.04}$ |

Table 13: Running time comparison of DreamMiss and other methods on three datasets.

| Methods | YooChoose | | KuaiRec | | Zhihu | |
|---|---|---|---|---|---|---|
| | Traning | Inferencing | Training | Inferencing | Training | Inferencing |
| SASRec | 01m 38s | 00m 06s | 02m 07s | 00m 08s | 00m 10s | 00m 01s |
| AdaRanker | 02m 29s | 00m 08s | 03m 38s | 00m 09s | 00m 14s | 00m 01s |
| DreamRec (DDPM) | 01m 31s | 21m 32s | 03m 59s | 32m 40s | 00m 14s | 01m 31s |
| DreamMiss (DDIM) | 01m 22s | 00m 13s | 02m 23s | 00m 23s | 00m 11s | 00m 01s |

shown in Table 13, DreamMiss substantially reduces the time cost during the inference phase than DreamRec and has a similar training time cost with other methods.

In large-scale recommendation systems, although the number of users and items increases, the sequence encoder (Transformer) and denoising models (MLP) we employ do not require excessive computational resources. Furthermore, Thompson sampling consumes almost no computational resources without involving model parameters. These factors enable DreamMiss to be effectively applied in large-scale systems. We detail some potential optimization strategies for larger datasets.

- Model Selection (Li et al., 2021): Choose lightweight architectures with lower computational complexity as the sequence encoder.
- Accelerating inference for DDMs: Reduce the iterative steps in the inference phase by consistency models (Song et al., 2023), knowledge distillation (Meng et al., 2023), or Reflow (Liu et al., 2023b).
- Starting Item (Lin et al., 2024): Instead of using pure noise as the starting item, use the mean of the sequence to reduce the iteration times during inference.

## G DISCUSSION ON THE MISSING DATA ISSUE

The missing data issue refers to the absence of behaviors in interaction sequences, which can lead to incomplete observed sequences in sequential recommendation tasks. For example, the complete sequence is $A \rightarrow B \rightarrow C \rightarrow D \rightarrow E \rightarrow F$. However, due to complex factors, we may only observe the partial interaction sequence $A \rightarrow C \rightarrow E \rightarrow F$ with $B$ and $D$ missing. Since the complete sequence is unavailable, the data missing in the observed sequence is uncertain and unpredictable. Such missing data issues can be roughly divided into two types: missing at random and missing not at random, as discussed in the related work A. Missing not at random often reflects bias, such as popularity bias and exposure bias. Missing at random is often caused by factors such as technical issues or privacy concerns.

DTS can effectively address the missing data issues when the observed sequence accurately reflects users' complete preferences, regardless of the specific factors (such as popularity bias, exposure bias, or technique issues). According to the probability models that reflect user preference evolution, DTS tends to remove items that have little impact on understanding users' original preference evolution, as shown in the consistent green curves in Figure B.2. By simulating uncertain missing data while maintaining the evolution of user preferences, DTS enables DDMs to generalize and effectively address the original missing data that does not disrupt users' complete preferences.

