# OpenReview forum: "Uncertainty-aware Guided Diffusion for Missing Data in Sequential Recommendation"
_ICLR.cc/2025/Conference — Submitted to ICLR 2025_

### Official Review · Reviewer_VkFH · 2024-10-30

**Soundness:** 3
**Presentation:** 3
**Contribution:** 2
**Rating:** 5
**Confidence:** 3

**Summary:**

This paper proposes a dual-side Thompson sampling (DTS) strategy to simulate the stochastical mechanism of missing data, and integrates it into denoising diffusion models for sequential recommendation. Extensive experimental results show that DreamMiss significantly outperforms baselines.

**Strengths:**

1. The experimental results are sound.
2. The paper is overall well written.

**Weaknesses:**

1. The missing data issue is not well discussed. Thompson sampling can not tackle all the missing data issues. How the proposed model can address which kind of missing data should be discussed in the paper.
2. The rationale of accelerated sampling is not well explained.
2. The hypothesis of stability scores is not reasonable. In recommendation model training, data are often shuffled to remove the dependency among samples. However, the stability scores seems the opposite way; it used the batch information for sampling. It contradicts with the traditional way.

**Questions:**

1. The line 7 of Algorithm 1 seems a bit different than other DDPM models. There's no square root above the $(1 - \alpha_{\tau_s})$. Is it a typo?
2. Can the authors explain why accelerated sampling works? It seems to reduce thousands of iterations to less than 100 rounds, which is a huge improvement.
3. Based on the performance of Dreasmiss, can we say the DreamRec has a overfitting problem? With much more rounds of iteration DreamRec has inferior performance.

---

> ### Author Response · Authors · 2024-11-22
> **Response to Reviewer VkFH --- Part 1/4**
>
> Dear Reviewer:
>
> We would like to express our sincere gratitude for your valuable feedback and
> constructive suggestions. We have detailed our responses **point-by-point** to your
> comments. We sincerely hope that our additional response has adequately **addressed
> your concerns**.
>
> ### Weakness 1:
> >**More discussion on the missing data issue:** Thompson sampling can not tackle all the missing data issues. **How** the proposed model can address **which kind of missing data** should be discussed in the paper.
>
> Thank you for your valuable feedback regarding the missing data issue. We acknowledge that Thompson sampling **does not address all types of missing data**. Below we **clarify how our proposed method specifically handles certain types of missing data**.
>
> - **Clarification for types of missing data issue:** The missing data issue refers to the absence of behaviors in interaction sequences, which can lead to incomplete observed sequences in sequential recommendation tasks.
> For example, the complete sequence is $A \rightarrow B \rightarrow C \rightarrow D \rightarrow E \rightarrow F$. However, due to complex factors, we may only **observe the partial interaction sequence $A\rightarrow C\rightarrow E \rightarrow F$ with $B$ and $D$ missing**.
>  Since the complete sequence is unavailable, the data missing in the observed sequence is **uncertain**. Such missing data issues can be roughly divided into two types: **missing at random** and **missing not at random**, as discussed **in the related work of our original submitted manuscript**.
>  Missing not at random often reflects bias, such as popularity bias and exposure bias. Missing at random is often caused by factors such as technical issues or privacy concerns.
> - **Clarification for missing data issue addressed by DTS:** DTS can effectively address the missing data issues **when the observed sequence accurately reflects users' complete preferences**, regardless of the specific factors (such as popularity bias, exposure bias, or technique issues). According to the probability models which reflect user preference evolution, DTS tends to remove items which have little impact on understanding users' original preference evolution, **as shown in the consistent green curves in Figure 1 (c) of our original submitted manuscript**. By simulating uncertain missing data while **maintaining the evolution of user preferences**, DTS enables DDMs to generalize and effectively address the original missing data that does not disrupt users' complete preferences.
>
> - **Experimental Validation:** To validate the effectiveness of DreamMiss in addressing the specific missing data issue, we construct two synthetic datasets: the `R` is constructed with missing data that occurs randomly and may not fully reflect user preferences; the `P` is designed with missing data aligned with user preferences. We conduct experiments on these two synthetic datasets, and the results are presented in Table R1.
>
> **Table R1:** Performance of DreamMiss and DreamRec on the two synthetic datasets.
> | Methods | YooChoose |  | KuaiRec |  | Zhihu |  |
> | :---: | :---: | :---: | :---: | :---: | :---: | :---: |
> |  | H@20(%) | N@20(%) | H@20(%) | N@20(%) | H@20(%) | N@20(%) |
> | DreamRec (R) | 5.11 | 2.16 | 4.99 | 4.16 | 1.95 | 0.68 |
> | DreamMiss (R) | 6.49 | 3.98 | 5.04 | 4.32 | 2.05 |0.69|
> | improv. (R) | 21.26% | 45.73% | 0.09% | 3.70% | 4.88% |1.45%|
> | DreamRec (P) |  5.15  | 2.17 | 4.93 | 4.01 | 2.08  | 0.70 |
> | DreamMiss (P) | 6.67 |4.12| 5.37 | 4.72| 2.21 | 0.76|
> | improv. (P) | 22.79% | 47.33% | 8.19% | 15.04% | 5.88% |7.89%|
>
> As shown in Table R1, DreamMiss is more effective on the "P" dataset, demonstrating a more significant improvement than on the dataset "R". This validates that DreamMiss can effectively address the missing data issue which do not change users' complete preferences.
>
> We will include this discussion in Appendix F.3 of our revision.

---

> ### Author Response · Authors · 2024-11-22
> **Response to Reviewer VkFH --- Part 2/4**
>
> ### Weakness 2 and  Question 2:
> >**More explanation for the rationale of accelerated sampling**
>
> Thank you for your valuable feedback! The accelerated sampling arises from DDIM, which reduces the number of iterative steps in the reverse process. Below, we provide a more detailed explanation of the rationale behind DDIM's accelerated sampling.
>
> - We first explain the different rationale of DDPM [1] and DDIM [2]. **For DDPM**, the forward process and reverse processes are both **Markovian chains**. The number of iterative steps in the reverse process is equal to that in the forward process **due to the reliance on the Markov assumption**. To ensure the generation quality, DDPM needs a substantial number of sampling steps in the reverse process. Unlike DDPM, **DDIM** accelerates the reverse process by **relaxing the Markov assumption**, achieving comparable generation quality with significantly fewer sampling steps. Below we list the difference between DDPM and DDIM in **Table R2**.
>
>
> **Table R2:** The rationale comparison for DDPM and DDIM.
> |                               | DDPM                                                         | DDIM                                                         |
> | ----------------------------- | ------------------------------------------------------------ | ------------------------------------------------------------ |
> | **Forward Process** | $q(x\_t\vert x\_0) = \mathcal{N}(x\_t;\sqrt{\bar \alpha\_t}x\_0,(1-\bar \alpha\_t)I)$ | $q(x\_t\vert x\_0) = \mathcal{N}(x\_t;\sqrt{\bar \alpha\_t}x\_0,(1-\bar \alpha\_t)I)$ |
> | **Reverse (Sampling) Process** | $q(\mathbf{x}^{t-1} \vert \mathbf{x}^t,\mathbf{x}^0)=\mathcal{N}(\mathbf{x}^{t-1};\tilde{\boldsymbol{\mu}}\_t(\mathbf{x}^t,\mathbf{x}^0),\tilde{\beta}\_t\mathbf{I})$,  where $\tilde{\boldsymbol{\mu}}\_t(\mathbf{x}^t,\mathbf{x}^0)=\frac{\sqrt{\bar{\alpha}\_{t-1}}\beta\_t}{1-\bar{\alpha}\_t}\mathbf{x}^0+\frac{\sqrt{\alpha\_t}(1-\bar{\alpha}\_t)}{1-\bar{\alpha}\_t}\mathbf{x}^t$, and $\tilde{\beta}\_t = \frac{1-\bar \alpha\_{t-1}}{1-\bar \alpha\_t}\beta\_t$ | $q\_\sigma(\boldsymbol{x}^{t-1}\vert \boldsymbol{x}^t,\boldsymbol{x}^0)=\mathcal{N}\left(\sqrt{\alpha\_{t-1}}\boldsymbol{x}^0+\sqrt{1-\alpha\_{t-1}-\sigma\_t^2}\cdot\frac{\boldsymbol{x}^t-\sqrt{\alpha\_t}\boldsymbol{x}^0}{\sqrt{1-\alpha\_t}},\sigma\_t^2\boldsymbol{I}\right)$, where $\sigma\_t = \eta \sqrt{\frac{1- \alpha\_{t-1}}{1- \alpha\_{t}}}\sqrt{1-\frac{ \alpha\_{t}}{ \alpha\_{t-1}}}$ |
> |**Training Objective**|  $L\_\gamma(\epsilon\_\theta):=\sum\_{t=1}^T\gamma\_t\mathbb{E}\_{\boldsymbol{x}^0\boldsymbol{\sim}q(\boldsymbol{x}^0),\epsilon\_t\boldsymbol{\sim}\mathcal{N}(\boldsymbol{0},\boldsymbol{I})}\left[\left\|\epsilon\_\theta^{(t)}(\sqrt{\alpha\_t}\boldsymbol{x}^0+\sqrt{1-\alpha\_t}\epsilon\_t)-\epsilon\_t\right\|\_2^2\right]$ |$L\_\gamma(\epsilon\_\theta):=\sum\_{t=1}^T\gamma\_t\mathbb{E}\_{\boldsymbol{x}^0\boldsymbol{\sim}q(\boldsymbol{x}^0),\epsilon\_t\boldsymbol{\sim}\mathcal{N}(\boldsymbol{0},\boldsymbol{I})}\left[\left\|\epsilon\_\theta^{(t)}(\sqrt{\alpha\_t}\boldsymbol{x}^0+\sqrt{1-\alpha\_t}\epsilon\_t)-\epsilon\_t\right\|\_2^2\right]$   |
> | **Sampling Steps**            | $\{1,...,T\}$                                                | $\{\tau\_1,...,\tau\_S\}\text{ which is a sub-sequence of }\{1,...,T\}$ |
>
> Both DDPM and DDIM are **trained under the same objective**, the difference lies in the reverse process of the inference phase.
> For DDIM, the $\sigma\_t = \eta \sqrt{\frac{1-\bar \alpha\_{t-1}}{1-\bar \alpha\_{t}}}\sqrt{1-\frac{ \bar \alpha\_{t}}{ \bar \alpha\_{t-1}}}$ is an arbitrary vector. When $\eta = 0$, the sampling process becomes a **deterministic process** as DDIM, and when $\eta = 1$, it corresponds to DDPM.
> Since the training objective **does not depend on the specific forward process** as long as $q\_\sigma(x\_t|x\_0)$ is fixed, **the deterministic sampling process of DDIM does not rely on the Markov chain** and can adopt a forward process with steps fewer than $T$. Thus the step-by-step sampling process in DDPM, which operates over $1\sim T$, can be modified to **iterate over a subset $\{\tau\_1,...,\tau\_S\}$** of $\{1,...,T\}$. This change enables the acceleration of the sampling process of DDIM.
>
> We will include this rationale explanation for DDIM's accelerating sampling in Appendix C of our revision.
>
> [1] Jonathan Ho, Ajay Jain, and Pieter Abbeel. Denoising diffusion probabilistic models. In NeurIPS, 2020a.
>
> [2] Jiaming Song, Chenlin Meng, and Stefano Ermon. Denoising diffusion implicit models. In ICLR. OpenReview.net, 2021a.

---

> ### Author Response · Authors · 2024-11-22
> **Response to Reviewer VkFH --- Part 3/4**
>
> ### Weakness 3:
> > **The reasonableness of stability scores** In recommendation model training, data are often shuffled to remove the dependency among samples. However, the stability scores seem the opposite way; it used the batch information for sampling. It contradicts the traditional way.
>
> Thank you for your insightful question regarding the handling of data dependencies in recommendation tasks. You are correct that dependency among samples is a common concern in many recommendation settings. Below we clarify the misunderstanding for the reasonableness of stability scores.
> - **Dataset Processing:** In our sequential recommendation tasks, the sequence samples are shuffled during batch loading, thereby eliminating dependencies among the samples in a batch.
> - **Stability score calculation:** The Stability scores are calculated by the entropy of item continuity scores within this sequence. For example, a sequence consists of 10 items, the normalized continuity scores for these items are $con=0.11,0.02,0.08,0.05,0.09,0.05,0.3,0.2,0.1$, and the sum of them is $1$. Then the stability score of this sequence is $h=-\sum_{n=1}^{n=9}con_n \log con_n\approx2.27$, representing the entropy of the continuity scores for this sequence. Greater uniformity in item continuity within a sequence results in higher entropy (indicating greater sequence stability), independent of other sequence samples. We normalize stability scores $h$ as $sta$ with softmax in a single batch, which **does not affect their relative values**; selecting sequences with relatively high stability in a batch **does not contradict traditional methods**.
> We will improve the clarification for the calculation of stability scores in our revision.
>
> ### Question 1:
> >**Typo:** The line 7 of Algorithm 1 lacks a square root.
>
> Thank you for reviewing our work so conscientiously! We sincerely apologize for the oversight and will carefully address the presentation issues in the revised version. Additionally, we will review the presentation throughout the entire manuscript.

---

> ### Author Response · Authors · 2024-11-22
> **Response to Reviewer VkFH --- Part 4/4**
>
> ### Question 3:
> > Based on the performance of DreasMiss, can we say that DreamRec has an overfitting problem? With more rounds of iteration, DreamRec has inferior performance.
>
> Thanks for your insightful question! We think it is **not an overfitting problem** for DreamRec. Below we clarify the misunderstanding for DreamRec's inferior performance.
>
> - We first explain that the more iterative steps in DreamRec come from the **DDPM's reverse process in the inference phase**, over that of **DDIM we employed in DreamMiss**, as shown in Table R3. Since they are trained similarly and there is no training or updating of model parameters in the inference phase, overfitting is not a concern. The inferior performance of DreamRec can be primarily attributed to its oversight of the missing data issue.
>
> **Table R3:** Running time comparison of DreamRec and DreamMiss per epoch.
> | Methods | YooChoose || KuaiRec || Zhihu ||
> |---------|-----------|-----------|-----------|-----------|-----------|-----------|
> |         | Training | Inferencing | Training | Inferencing | Training | Inferencing |
> | DreamRec (DDPM) | 01m 31s | 21m 32s | 03m 59s  | 32m 40s     | 00m 14s  | 01m 31s    |
> | DreamMiss (DDIM) | 01m 22s | 00m 13s | 02m 23s  | 00m 11s     | 00m 11s  | 00m 01s    |
>
>
> - Then we **conduct additional experiments to demonstrate that it is the DTS rather than DDIM that brings performance improvement for DreamRec**. Specifically, we apply DDPM and DDIM for DreamRec and DreamMiss respectively to validate their impact on the performance of diffusion-based recommenders. The experimental results are presented **in Table R4**.
>
>
> **Table R4:** Performance comparison of DDPM and DDIM for diffusion-based recommenders.
> | Methods | YooChoose |  | KuaiRec |  | Zhihu |  |
> | :---: | :---: | :---: | :---: | :---: | :---: | :---: |
> |  | H@20(%) | N@20(%) | H@20(%) | N@20(%) | H@20(%) | N@20(%) |
> | DreamRec (DDPM)| 4.78  | 2.23 | 5.16  | 4.11  | 2.26 | 0.79  |
> | DreamRec (DDIM)| 4.85    | 2.26 | 4.93  | 4.01 | 2.25 |  0.82  |
> | DreamMiss (DDPM)| 6.82  | 4.33 | 5.49  | 4.74  | 2.45  | 0.85 |
> | DreamMiss (DDIM) | 6.90 |4.34 | 5.48 | 4.77| 2.65 | 0.88  |
>
> As shown in Table R4, the performance of DDPM and DDIM is comparable, as evidenced by the similar results for DreamRec (DDPM) and DreamRec (DDIM). This validates that **the rounds of iteration in the inference phase will not cause performance improvement**. Furthermore, DreamMiss consistently outperforms DreamRec, demonstrating the effectiveness of our DTS in enhancing the robustness of diffusion models. **Thus it is the DTS that results in performance improvement instead of the accelerated sampling with fewer iterative rounds in DDIM.**
>
>
> - We believe that the inferior performance of DreamRec stems from **training solely on the observed sequence**, struggling with complex missing data. The DTS of DreamMiss generates uncertainty-aware missing data, enhancing the robustness of diffusion models in real-world scenarios through consistency regularization.
>
>
> We sincerely hope that our additional response has adequately **addressed your concerns**. If so, we would greatly appreciate your consideration in **raising the score**. If there are any remaining concerns, **please let us know**, and we will **continue to actively address your comments** and **improve our work**.

---

> ### Author Response · Authors · 2024-11-25
>
> Dear Reviewer VkFH,
>
> We deeply appreciate your valuable feedback and the time you've taken to review our work with these insights, especially during this busy period.
>
> Regarding your primary concern about the **clarification for missing data issue**, we have clarified the missing data issue that DTS can address with additional experiments. For the point you raised about the **rationale of accelerated sampling for DDIM**, we have provided a more detailed explanation of the rationale behind DDIM's accelerated sampling and compared it with DDPM. Concerning **reasonability of stability scores**, we have clarified the calculation of stability scores, which are based on item continuity scores within a sequence and align with traditional methods.
>
> We hope that our responses have addressed most of your concerns. If so, we kindly ask if you could consider revising the overall rating of our paper. If there are any remaining concerns, we would be more than happy to further discuss them with you, especially since the rebuttal period is nearing its end. Please let us know if you have any additional questions.
>
> Thank you once again for your support and understanding!
>
> Best regards,
>
> Authors

---

> ### Author Response · Authors · 2024-11-30
> **Kind Reminder**
>
> Dear Reviewer VkFH,
>
> As the discussion phase is coming to an end, we sincerely hope to address your concerns and engage in further discussion. If you have any remaining questions, please feel free to reach out directly. We would like to take this final opportunity to improve our work.
>
> Best regards,
>
> Authors of Paper 7669

---

> ### Author Response · Authors · 2024-12-03
> **Looking forward to your feedback**
>
> Dear Reviewer VkFH,
>
> As the discussion phase is coming to an end, we sincerely hope to address your concerns thoroughly. If you find our responses satisfactory, we would greatly appreciate your consideration of revising your score. Thank you once again for your valuable feedback and thoughtful suggestions.
>
> Best regards,
>
> Authors of Paper 7669

---

### Official Review · Reviewer_xLHr · 2024-10-31

**Soundness:** 4
**Presentation:** 4
**Contribution:** 4
**Rating:** 10
**Confidence:** 1

**Summary:**

Sorry, I am not familiar with this research area so I can't give a valuable score. Please overlook my score.

**Strengths:**

Sorry, I am not familiar with this research area so I can't give a valuable score. Please overlook my score.

**Weaknesses:**

Sorry, I am not familiar with this research area so I can't give a valuable score. Please overlook my score.

**Questions:**

Sorry, I am not familiar with this research area so I can't give a valuable score. Please overlook my score.

---

> ### Author Response · Authors · 2024-11-22
> **Response to Reviewer xLHr**
>
> Dear Reviewer:
>
> Thank you for your feedback and for the score you provided. We appreciate your honesty regarding your familiarity with this research area. We have made revisions and improvements to the content in response to the comments received. We hope these changes enhance the clarity of our work and assist in your understanding of the paper.
>
> We would be grateful for any additional insights or suggestions you might have, as your perspective is valuable to us.
>
> Thank you once again for your time and consideration.

---

### Official Review · Reviewer_fpPx · 2024-11-03

**Soundness:** 2
**Presentation:** 2
**Contribution:** 2
**Rating:** 5
**Confidence:** 4

**Summary:**

The paper proposes DreamMiss, a novel approach to handle missing data in sequential recommendation systems using denoising diffusion models (DDMs). The key innovation is a dual-side Thompson sampling (DTS) strategy that simulates missing data stochastically while preserving user preference patterns. The approach uses two probability models: 1) A local model that captures continuity between adjacent items A global model that evaluates sequence stability. 2) DreamMiss is implemented using denoising diffusion implicit models (DDIM) for faster sampling and achieves consistency regularization through uncertainty-aware guidance.

**Strengths:**

The paper is generally well-written.
The problem of missing data is crucial for developing sequential recommenders.

**Weaknesses:**

W1. Conceptual Clarity and Methodology Concerns.
The paper lacks clear theoretical justification for how generating additional missing data helps address the original missing data problem.
The approach of simulating missing data from already incomplete sequences raises questions about potential error propagation.
The methodology appears counterintuitive compared to traditional approaches that aim to recover or compensate for missing data.
The paper would benefit from a more rigorous theoretical analysis of why this approach is superior to data completion methods.

W2. Limited Validation of Dual-side Thompson Sampling (DTS).
The paper does not sufficiently justify why DTS is specifically effective for diffusion-based sequential recommenders.
There is inadequate theoretical analysis or empirical validation of the reliability of the continuity and stability metrics.
The generalizability of DTS to other recommendation architectures needs more thorough investigation.
The robustness of the probability models used in DTS requires more comprehensive validation.

W3. Incomplete Baseline Comparisons.
Notable omissions of important state-of-the-art baselines, particularly:
[1] Lin, Y.; Wang, C.; Chen, Z.; Ren, Z.; Xin, X.; Yan, Q.; de Rijke, M.; Cheng, X.; and Ren, P. A Self-Correcting Sequential Recommender. In TheWebConf 2023.
[2] Zhang, C.; Han, Q.; Chen, R.; Zhao, X.; Tang, P.; and Song, H. SSDRec: Self-Augmented Sequence Denoising
for Sequential Recommendation. In ICDE 2024.
These omissions make it difficult to fully assess the comparative advantages of the proposed method.
The evaluation would be more convincing with a more comprehensive comparison against recent approaches.

W4. Scalability Limitations.
The computational complexity of the proposed method may limit its practical applications.
Insufficient discussion of performance on large-scale recommendation systems.
Limited analysis of computational resource requirements for real-world deployment.
Need for more detailed discussion of potential optimization strategies for larger datasets.

W5. Dataset Limitations.
The evaluation relies on relatively small-scale datasets.
Questions about generalizability to larger, more complex real-world recommendation scenarios.
Need for validation on more diverse and larger-scale datasets.
Limited demonstration of effectiveness across different domains and data distributions.

**Questions:**

Please refer to the Weaknesses.

---

> ### Author Response · Authors · 2024-11-22
> **Response to Reviewer fpPx --- Part 1/10**
>
> Dear Reviewer:
>
> We would like to express our sincere gratitude for your valuable feedback and constructive suggestions. We have detailed our responses **point-by-point** to address your concerns.
>
> ### Weakness 1
>
> > **Conceptual Clarity and Methodology Concerns:** Clear theoretical justification for how generating additional missing data helps address the original missing data problem and why this approach is superior to data completion methods are needed.
>
> Thank you for your valuable feedback, which is of great help in improving the clarity of our method. Formally, we denote the original missing data as $\delta$, the observed sequence as $\mathbf{e}\_{1:N-1}$, the simulated missing data as $\delta'$, the complete sequence as $\mathbf{e}\_{1:N-1}\oplus \delta$, and the simulated sequence as $\mathbf{e}\_{1:N-1}\ominus \delta'$
>
>
> - We first explain **why we did not adopt completion methods:**
>   -  **Unavailability of complete sequence:** Due to the lack of prior knowledge for the complete sequence $\mathbf{e}\_{1:N-1}\oplus \delta$ **from an omniscient perspective**, we cannot pinpoint where the missing data $\delta$ occurs or what specific content it pertains to.
>   - **Uncertainty of missing data:** Due to the variability of user preferences, the missing data $\delta$ is **uncertain** in the observed interaction sequence $\mathbf{e}\_{1:N-1}$. User interaction may miss anywhere **stochastically** in the observed sequence.
>
>
> Thus, compensating for missing data may **introduce additional errors as shown in Figure 1 (a) of our original submitted manuscript**. The unavailability of complete sequence $\mathbf{e}\_{1:N-1}\oplus\delta$  and the uncertainty of missing data motivates us to generate additional missing data $\delta'$ **stochastically** from the observed sequence $\mathbf{e}\_{1:N-1}$, treating $\mathbf{e}\_{1:N-1}$ as the **complete sequence** for the simulated sequence $\mathbf{e}\_{1:N-1} \ominus \delta'$.

---

> ### Author Response · Authors · 2024-11-22
> **Response to Reviewer fpPx --- Part 2/10**
>
> - Then we justify **how generating additional missing data helps address the original missing data problem** through **extrapolation** [1] and **consistency regularization** [2].
>
>   - **Extrapolation:**
> Let $\hat{\mathbf{g}}$, $\bar{\mathbf{g}}$, and $\tilde{\mathbf{g}}$ denote the guidance encoded from observed sequence $\mathbf{e}\_{1:N-1}$, complete sequence $\mathbf{e}\_{1:N-1}\oplus \delta$ and simulated sequence $\mathbf{e}\_{1:N-1}\ominus \delta'$, respectively. Our objective is to demonstrate the validity of the extrapolation, specifically the inequality $\|f\_\theta(\mathbf{e}^{\tau\_s}\_N,\bar{\mathbf{g}},\tau\_s)-f\_\theta(\mathbf{e}^{\tau\_s}\_N,\hat{\mathbf{g}},\tau\_s)\|\le C\|f\_\theta(\mathbf{e}^{\tau\_s}\_N,\hat{\mathbf{g}},\tau\_s)-f\_\theta(\mathbf{e}^{\tau\_s}\_N,\tilde{\mathbf{g}},\tau\_s)\|$ for some constant $C$, where $\mathbf{e}^{\tau\_s}\_N$ represents the next interacted item with noise of $\tau\_s$ time steps, $f\_\theta$ is the denoising model.
> Applying Taylor's Formula, we can express the two functions as follows:
>
> \begin{aligned}
> f\_\theta(\mathbf{e}^{\tau\_s}\_{N},\bar{\mathbf{g}},\tau\_s)=f\_\theta(\mathbf{e}^{\tau\_s}\_{N},\hat{\mathbf{g}},\tau\_s)+(\bar{\mathbf{g}}-\hat{\mathbf{g}})^\top\nabla\_g f\_\theta(\mathbf{e}^{\tau\_s}\_{N},\hat{\mathbf{g}},\tau\_s)+o(\|\bar{\mathbf{g}}-\hat{\mathbf{g}}\|),\\\f\_\theta(\mathbf{e}^{\tau\_s}\_{N},\tilde{\mathbf{g}},\tau\_s)=f\_\theta(\mathbf{e}^{\tau\_s}\_{N},\hat{\mathbf{g}},\tau\_s)+(\tilde{\mathbf{g}}-\hat{\mathbf{g}})^\top\nabla\_g f\_\theta(\mathbf{e}^{\tau\_s}\_{N},\hat{\mathbf{g}},\tau\_s)+o(\|\tilde{\mathbf{g}}-\hat{\mathbf{g}}\|).
> \end{aligned}
>
> Then, combining the above two equalities, we have
>
> \begin{aligned}&f\_\theta(\mathbf{e}^{\tau\_s}\_{N},\bar{\mathbf{g}},\tau\_s)-f\_\theta(\mathbf{e}^{\tau\_s}\_{N},\hat{\mathbf{g}},\tau\_s)\\\\&=(\bar{\mathbf{g}}-\hat{\mathbf{g}})^\top\nabla f\_\theta(\mathbf{e}^{\tau\_s}\_{N},\hat{\mathbf{g}},\tau\_s)+o(\|\bar{\mathbf{g}}-\hat{\mathbf{g}}\|)\\\\&=\frac{(\bar{\mathbf{g}}-\hat{\mathbf{g}})^\top\nabla\_g f\_\theta(\mathbf{e}^{\tau\_s}\_{N},\hat{\mathbf{g}},\tau\_s)}{(\tilde{\mathbf{g}}-\hat{\mathbf{g}})^\top\nabla\_g f\_\theta(\mathbf{e}^{\tau\_s}\_{N},\hat{\mathbf{g}},\tau\_s)}(\tilde{\mathbf{g}}-\hat{\mathbf{g}})^\top\nabla\_g f\_\theta(\mathbf{e}^{\tau\_s}\_{N},\hat{\mathbf{g}},\tau\_s)+o(\|\bar{\mathbf{g}}-\hat{\mathbf{g}}\|)\\\\&=\frac{(\bar{\mathbf{g}}-\hat{\mathbf{g}})^\top\nabla\_g f\_\theta(\mathbf{e}^{\tau\_s}\_{N},\hat{\mathbf{g}},\tau\_s)}{(\tilde{\mathbf{g}}-\hat{\mathbf{g}})^\top\nabla\_g f\_\theta(\mathbf{e}^{\tau\_s}\_{N},\hat{\mathbf{g}},\tau\_s)}(f\_\theta(\mathbf{e}^{\tau\_s}\_{N},\tilde{\mathbf{g}},\tau\_s)-f\_\theta(\mathbf{e}^{\tau\_s}\_{N},\hat{\mathbf{g}},\tau\_s))+o(\|\bar{\mathbf{g}}-\hat{\mathbf{g}}\|)+o(\|\tilde{\mathbf{g}}-\hat{\mathbf{g}}\|),\end{aligned}
>
> where we assume that $(\tilde{\mathbf{g}}-\hat{\mathbf{g}})^\top\nabla\_gf\_\theta(\mathbf{e}^{\tau\_s}\_{N},\hat{\mathbf{g}},\tau\_s)\neq 0$. Thus, we obtain the following inequality:
>
> \begin{aligned}&\|f\_\theta(\mathbf{e}^{\tau\_s}\_{N},\bar{\mathbf{g}},\tau\_s)-f\_\theta(\mathbf{e}^{\tau\_s}\_{N},\hat{\mathbf{g}},\tau\_s)\|\\\\&\le\left|\frac{(\bar{\mathbf{g}}-\hat{\mathbf{g}})^\top\nabla\_g f\_\theta(\mathbf{e}^{\tau\_s}\_{N},\hat{\mathbf{g}},\tau\_s)}{(\tilde{\mathbf{g}}-\hat{\mathbf{g}})^\top\nabla\_g f\_\theta(\mathbf{e}^{\tau\_s}\_{N},\hat{\mathbf{g}},\tau\_s)}\right|\|f\_\theta(\mathbf{e}^{\tau\_s}\_{N},\hat{\mathbf{g}},\tau\_s)-f\_\theta(\mathbf{e}^{\tau\_s}\_{N},\tilde{\mathbf{g}},\tau\_s)\|+o(\|\bar{\mathbf{g}}-\hat{\mathbf{g}}\|)+o(\|\tilde{\mathbf{g}}-\hat{\mathbf{g}}\|),\end{aligned}
>
>  where $\|f\_\theta(\mathbf{e}^{\tau\_s}\_{N},\hat{\mathbf{g}},\tau\_s)-f\_\theta(\mathbf{e}^{\tau\_s}\_{N},\tilde{\mathbf{g}},\tau\_s)\|$ is the distance between the prediction from observed sequence and simulated sequence. To bound the coefficients $\left|\frac{(\bar{\mathbf{g}}-\hat{\mathbf{g}})^\top\nabla\_g f\_\theta(\mathbf{e}^{\tau\_s}\_{N},\hat{\mathbf{g}},\tau\_s)}{(\tilde{\mathbf{g}}-\hat{\mathbf{g}})^\top\nabla\_g f\_\theta(\mathbf{e}^{\tau\_s}\_{N},\hat{\mathbf{g}},\tau\_s)}\right|$, we need to analyze the two inner products, ${(\bar{\mathbf{g}}-\hat{\mathbf{g}})^\top\nabla\_g f\_\theta(\mathbf{e}^{\tau\_s}\_{N},\hat{\mathbf{g}},\tau\_s)}$ and ${(\tilde{\mathbf{g}}-\hat{\mathbf{g}})^\top\nabla\_g f\_\theta(\mathbf{e}^{\tau\_s}\_{N},\hat{\mathbf{g}},\tau\_s)}$. If the simulated missing data process from $\mathbf{e}\_{1:N-1}$ to $\mathbf{e}\_{1:N-1}\ominus \delta'$ can align well with that from $\mathbf{e}\_{1:N-1}\oplus\delta$ to $\mathbf{e}\_{1:N-1}$---the difference between the two pair of data will be roughly equivalent. Consequently, the two differences in guidance, $\bar{\mathbf{g}}-\hat{\mathbf{g}}$ and $\hat{\mathbf{g}}-\tilde{\mathbf{g}}$, will also be approximately equal. In this scenario, the coefficient will be close to 1, resulting in a bounded value $C>0$.

---

> ### Author Response · Authors · 2024-11-22
> **Response to Reviewer fpPx --- Part 3/10**
>
> - **Consistency regularization:** To ensure the effectiveness of extrapolation, we leverage consistency regularization to minimize $\|f\_\theta(\mathbf{e}^{\tau\_s}\_{N},\hat{\mathbf{g}},\tau\_s)-f\_\theta(\mathbf{e}^{\tau\_s}\_{N},\tilde{\mathbf{g}},\tau\_s)\|$. Since the interaction sequences are edited stochastically across different epochs, $\hat{\mathbf{g}}$ and $\tilde{\mathbf{g}}$ can serve as the perturbated pairs. Let the ground-truth label of the next item be $\mathbf{e}\_N^0$.
>    By completing the square, we obtain the inequality:
>    $||f\_\theta(\mathbf{e}\_N^{\tau\_s},\hat{\mathbf{g}}, \tau\_{s})-f\_\theta(\mathbf{e}\_N^{\tau\_s},\tilde{\mathbf{g}}, \tau\_{s})||\_2^2 \leq  2\left(||f\_\theta(\mathbf{e}\_N^{\tau\_s},\hat{\mathbf{g}},\tau\_{s})-\mathbf{e}\_N^0||\_2^2+||f\_\theta(\mathbf{e}\_N^{\tau\_s},\tilde{\mathbf{g}}, \tau\_{s})-\mathbf{e}\_N^0||\_2^2\right)$. Consequently, we can achieve consistency regularization of minimizing the left-hand side by minimizing the right-hand side, which stems from our reconstruction loss.
>
>   We improve **the explanation in section 3.3 of revision**, and include more explanation for our motivation and method **in Appendix B.1 and B.3 of our revision**.
>
> [1] David Krueger, Ethan Caballero, Jörn-Henrik Jacobsen, Amy Zhang, Jonathan Binas, Dinghuai Zhang, Rémi Le Priol, and Aaron C. Courville. Out-of-distribution generalization via risk extrapolation (rex). In ICML, volume 139 of Proceedings of Machine Learning Research, pp. 5815–5826. PMLR, 2021.
>
> [2] Han Zhang, Zizhao Zhang, Augustus Odena, and Honglak Lee. Consistency regularization for
> generative adversarial networks. In ICLR. OpenReview.net, 2020.
>
>
> - Finally, we **conduct experiments** to validate the superiority of DreamMiss over data completion methods, we make a comparison for **the three methods (i.e., simulating missing data, leveraging observed data, and recovering missing data)** on the missing data scenarios.
>   - **In Table 1 of our original submitted manuscript**, we have compared DreamMiss with some **imputation-based baseline methods (DiffuASR, CaDiRec, and PDRec)**. The superior performance of DreamMiss demonstrates the effectiveness of simulating missing data.
>   - Additionally, following the setting of **section 5.4 of our original submitted manuscript**, we create synthetic datasets with missing data proportions of 10%, 20%, and 30%. We **compare the three methods on the synthetic datasets** and the detailed results are shown in **Table R1**.
>
>
>     **Table R1:** Performance comparison for the three methods (i.e., simulating missing data with DreamMiss, leveraging observed data with DreamRec, and recovering missing data with DreamImp).
>     | Methods | YooChoose |  | KuaiRec |  | Zhihu |  |
>     | :---: | :---: | :---: | :---: | :---: | :---: | :---: |
>     |  | H@20(%) | N@20(%) | H@20(%) | N@20(%) | H@20(%) | N@20(%) |
>     | DreamRec (10%) | 5.11 | 2.16 | 4.99 | 4.16 | 1.95 | 0.68 |
>     | DreamImp (10%) | 5.32 | 2.28 | 4.89 | 4.07 | 1.98 | 0.68 |
>     | DreamMiss (10%) | **6.49** | **3.98** | **5.04** | **4.32** | **2.05**|**0.69**|
>     | DreamRec (20%) | 4.87 | 2.03 | 4.59 | 3.81 | 1.89 | 0.68 |
>     | DreamImp (20%) | 4.91 | 2.55 | 4.65 | 3.76 | 1.91 | 0.70 |
>     | DreamMiss (20%) | **6.26**|**3.66** | **4.77**| **4.07**| **2.22** | **0.71**|
>     | DreamRec (30%) | 4.74 | 1.97 | 4.18 | 3.48 | 1.89 | 0.67 |
>     | DreamImp (30%) | 4.73 | 2.18 | 4.21 | 3.59 | 1.93 | 0.68 |
>     | DreamMiss (30%) | **5.92**| **3.29**| **4.49**| **3.69** | **2.12** | **0.70**|
>
>     DreamImp is our imputation-based algorithm designed for diffusion models. In the absence of prior knowledge about the complete sequence, we use the average of two adjacent items for imputation at random positions. Experimental results indicate that the method of simulating missing data (DreamMiss) outperforms approaches that rely solely on observed data (DreamRec) or attempt to recover missing data (DreamImp). We **include the experiments in Appendix F.3 of our revision**.

---

> ### Author Response · Authors · 2024-11-22
> **Response to Reviewer fpPx --- Part 4/10**
>
> ### Weakness 2
>
> > **Limited Validation of Dual-side Thompson Sampling (DTS):** The reason for DTS is specifically effective for **diffusion-based sequential recommenders** should be justified. Theoretical analysis or empirical validation of the **reliability of the continuity and stability metrics** should be provided. **The generalizability of DTS to other recommendation architectures** needs a more thorough investigation. **The robustness of the probability models** used in DTS requires more comprehensive validation.
>
> Thanks for your valuable suggestions! We would conduct additional experiments and make a deeper analysis of the Dual-side Thompson Sampling (DTS).
>
>  - First, we explain why **DTS** is specifically effective for **diffusion-based sequential recommenders**.
>    - **For diffusion-based recommenders:** Introducing missing data with DTS can be considered as **introducing a form of noise within the sequence**, which can enhance DDMs' denoising ability to missing data rather than Gaussian noise. Additionally, DTS can be a **general algorithm to enhance various recommendation architectures' robustness** against uncertain missing data, by simulating missing data without disrupting user preference evolution.
>    - **For DTS strategy:** The advantage of DTS lies in that it can simulate missing data **stochastically without disrupting user preference**. By balancing the exploration and exploitation based on the known user preference in probability models, DTS is more likely to remove behavior that has little impact on understanding user preference evolution, as evidenced by the consistent green curves between the observed sequence and the one edited by DTS in Figure 1 of our original submitted manuscript.
>
>  - Next, we conduct additional experiments to validate the **generalizability of DTS to different recommendation architectures**. In addition to the experimental results **in Table 4 of our original submitted manuscript (generalizing DTS to GRU4Rec, SASRec, CL4Rec)**, we conduct further experiments to assess the generalizability of DTS across **other recommendation architectures (Caser [1], AdaRanker [2], and DiffuASR [3])**. The results of these experiments are presented in **Table R2**.
>
>     **Table R2:** Experimental results for the generalizability of DTS across **other recommendation architectures**.
>
>     | Methods | YooChoose |  | KuaiRec |  | Zhihu |  |
>     | :---: | :---: | :---: | :---: | :---: | :---: | :---: |
>     |  | H@20(%) | N@20 (%) | H@20 (%) | N@20(%) | H@20(%) | N@20(%) |
>     | **GRU4Rec** | 3.89 | 1.62 | 3.32 | 1.23 | 1.78 | 0.67 |
>     | + DTS | 3.96 | 1.72 | 3.43 | 1.40 | 1.83 | 0.68 |
>     | improv. | 1.77% | 5.81% | 3.21% | 12.14% | 2.73% | 1.47% |
>     | **SASRec** | 3.68 | 1.63 | 3.92 | 1.53 | 1.62 | 0.60 |
>     | +DTS | 3.98 | 1.58 | 3.96 | 1.63 | 1.70 | 0.72 |
>     | improv. | 7.54% | -3.07% | 1.01% | 6.13% | 4.71% |**11.11%** |
>     | **CL4Rec** | 4.45 | 1.86 | 4.25 | 2.01 | 2.03 | 0.74 |
>     | +DTS | 4.63 | 1.88 | 4.57 | 2.25 | 2.16 | 0.78 |
>     | improv. | 3.89% | 1.06% | 7.00% | 10.67% | 6.02% | 5.13% |
>     | **Caser** | 4.06 | 1.88 | 2.88 | 1.07 | 1.57 | 0.59 |
>     | +DTS | 4.27 | 2.01 | 3.19  |  1.09 | 1.72 | 0.65 |
>     | improv. | 4.92% | 6.47% | **9.72%** | 1.83% | 8.72% | 9.23% |
>     | **AdaRanker** | 3.74 | 1.67 | 4.14 | 1.89 | 1.70 | 0.61 |
>     | +DTS | 4.16 | 1.97 | 4.33  |1.93 | 1.64 | 0.67 |
>     | improv. | 10.10% | 15.22% | 4.39% | 2.07% | -3.53% | 8.96% |
>     | **DiffuASR** | 4.48 | 1.92 | 4.53 | 3.30 | 2.05 | 0.71 |
>     | +DTS | 4.66 | 2.08 | 4.58  | 3.98 | 2.05 | 0.73 |
>     | improv. | 3.86% | 7.69% | 1.09% | **17.09%** | 0.00% | 2.82% |
>     | **DreamRec** | 4.78 | 2.23 | 5.16 | 4.11 | 2.26 | 0.79 |
>     | +DTS | 6.90 | 4.34 | 5.48 | 4.77 | 2.65 | 0.88 |
>     | improv. | **30.72%** | **48.62%** | 5.84% | 13.84% | **14.72%** | 10.23% |
>
>     The experimental results indicate that applying DTS across various recommendation architectures leads to consistent improvements in recommendation performance, demonstrating that DTS is a general algorithm for enhancing the robustness of recommendation systems. Furthermore, the significant performance gains observed with DTS in DreamRec underscore its effectiveness for diffusion-based models (DDMs), **owing to their capability in modeling complex data distributions and denoising the missing data**.
>
>     [1] Jiaxi Tang and Ke Wang. Personalized top-n sequential recommendation via convolutional sequence embedding. In WSDM, pp. 565–573. ACM, 2018.
>
>     [2] Xinyan Fan, Jianxun Lian, Wayne Xin Zhao, Zheng Liu, Chaozhuo Li, and Xing Xie. Ada-ranker: A data distribution adaptive ranking paradigm for sequential recommendation. In SIGIR, pp.1599–1610. ACM, 2022a
>
>     [3] Qidong Liu, Fan Yan, Xiangyu Zhao, Zhaocheng Du, Huifeng Guo, Ruiming Tang, and Feng Tian. Diffusion augmentation for sequential recommendation. In CIKM, pp. 1576–1586. ACM, 2023.

---

> ### Author Response · Authors · 2024-11-22
> **Response to Reviewer fpPx --- Part 5/10**
>
> - Then we **conduct experiments** to verify the **advantages of the Thompson sampling strategy compared to other sampling strategies**, including random-based sampling (sample uniformly without exploiting known information), time-based sampling (sample based on timestamps), and model-based sampling (sample based on another model while lacking supervision signals). More details are introduced in Appendix B.2 of our revision. The experimental results are presented **in Table R3**. The `Base` represents the diffusion-based recommender without a sampling strategy.
>
>     **Table R3:** The performance comparison of **different sampling strategies** for uncertainty-aware guided diffusion for recommendation.
>
>     | Methods | YooChoose |  | KuaiRec |  | Zhihu |  |
>     | :---: | :---: | :---: | :---: | :---: | :---: | :---: |
>     |  | H@20(%) | N@20(%) | H@20(%) | N@20(%) | H@20(%) | N@20(%) |
>     | Base | 4.78  | 2.23| 5.16  | 4.11| 2.26 | 0.79|
>     | Random-based | 6.24  | 3.91  | 5.37  | 4.19 | 2.30  | 0.80  |
>     | Time-based| 6.28  | 3.96 | 5.20 | 4.55 | 2.24  | 0.79 |
>     | Model-based | 6.49  | 3.62  | 5.18  |  4.59  | 2.36| 0.80  |
>     | DreamMiss | **6.90**|  **4.34** | **5.48** |  **4.77** | **2.65** | **0. 88** |
>
>     The experimental results in **Table R3** indicate that different sampling strategies have a noticeable impact on the diffusion-based recommender compared to the `Base` approach. However, DreamMiss demonstrates superior performance than other sampling strategies, **confirming the advantages of DTS**.

---

> ### Author Response · Authors · 2024-11-22
> **Response to Reviewer fpPx --- Part 6/10**
>
> - Finally, we conduct experiments to validate the **reliability of the continuity and stability metrics** of probability models.
>   - Notably, there is **a gap in the existing research on metrics for measuring the observed sequences**. One of our contributions is the proposal of two new metrics: local item continuity and global sequence stability.
>
>   - To demonstrate their effectiveness, we have conducted **an ablation study with experimental results presented in Table 2 of our original submitted manuscript**. This study assesses the recommendation performance **with and without continuity and stability metrics**.
>
>   - Here, in addition to our original ablation study, we **propose other metrics** and carry out further experiments to compare their performance. Specifically, we introduce additional metrics including item popularity, item diversity within a sequence, item position in the sequence, and sequence length for the probability models. The results of these experiments are displayed in **Table R4**.
>
>     **Table R4:** Performance comparison of different metrics for probability models.
>     | Methods | YooChoose |  | KuaiRec |  | Zhihu |  |
>     | :---: | :---: | :---: | :---: | :---: | :---: | :---: |
>     |  | H@20(%) | N@20(%) | H@20(%) | N@20(%) | H@20(%) | N@20(%) |
>     | Base | 4.78  | 2.23 | 5.16  | 4.11 | 2.26  | 0.79 |
>     | w/o GL| 6.24 | 3.91| 5.37 | 4.19  | 2.30 | 0.80 |
>     | w/o L | 6.41 | 4.26 | 5.44 | 4.63 | 2.34 | 0.86  |
>     | w/o G | 6.48  | 4.29 | 5.43 | 4.64  | 2.44  | 0.81 |
>     | w Popularity| 6.28 | 4.18 | 5.46   | 4.57 | 2.38  | 0.80  |
>     | w Diversity | 6.26 | 4.20 | 5.45 | 4.52  | 2.29 | 0.81 |
>     | w Seq_len | 6.27 | 4.30 | 5.46| 4.54  | 2.30   | 0.83  |
>     | w Item_pos| 6.28  | 3.96 | 5.20| 4.55 | 2.24 | 0.79  |
>     | DreamMiss | 6.90  |4.34 | 5.48 | 4.77 | 2.65| 0.88  |
>
>   Specifically, 'w/o G', 'w/o L', and 'w/o GL' represent the absence of global sequence stability metric, local item continuity metric, or both, respectively. The metric of item popularity is measured by the frequency of each item in all interactions, which reflects **popularity bias**. The metric of item diversity within a sequence is calculated to indicate the diversity of user preferences. The metric of sequence length serves as a measure of user activity, while the metric of item position reflects short-term and long-term preferences. As shown in Table R3, `w/o G`, `w/o L`, `w Popularity`, `w Diversity`, `w Seq-len`, and `w Item-pos` outperform the `Base` and `w/o GL`, highlighting the **robustness of probability models** in helping to alleviate the missing data issue. Moreover, the superior performance of DreamMiss **demonstrates the reliability of continuity and stability metrics for probability models**.
>
> - To further validate the **robustness of the probability models**, we have conducted experiments over five experimental runs with **different random seeds** and on **different datasets**, reporting the average performance with their corresponding standard deviations in the tables of our submission. The experimental results are **relatively stable** as the standard deviations of DreamMiss are similar to those of baselines. Additionally, we validate the robustness of DreamMiss to the proportion of missing data in Table 8 of our revision and the sensitivity of DreamMiss to the threshold of simulation in Appendix F.2 of our revision. Experimental results demonstrate the robustness of probability models.
>
> **We include the explanation and experiments in Sections 4.3 and Appendix B.2, F.4, and F.5 in our revision.**

---

> ### Author Response · Authors · 2024-11-22
> **Response to Reviewer fpPx --- Part 7/10**
>
> ### Weakness 3
> > **Incomplete Baseline Comparisons:**  [1] STEAM: A Self-Correcting Sequential Recommender.
> [2] SSDRec: Self-Augmented Sequence Denoising for Sequential Recommendation.
> Thanks for your valuable suggestion! Based on your suggestions, we have made additional experiments on these baseline methods.
>
> Thank you for your valuable suggestion! We conduct additional experiments to implement the two baselines on our three datasets (YooChoose, KuaiRec, and Zhihu) and we include the results in our main experiments of revision.
>
> **Table R5:**  Performance of additional baseline methods.
> | Methods | YooChoose |  | KuaiRec |  | Zhihu |  |
> | :---: | :---: | :---: | :---: | :---: | :---: | :---: |
> |  | H@20(%) | N@20(%) | H@20(%) | N@20(%) | H@20(%) | N@20(%) |
> | SASRec| 3.68  | 1.63 | 3.92  | 1.53  | 1.62 | 0.60 |
> | STEAM| 4.69   | 1.76 | 4.98 | 2.90  | 1.75 | 0.69  |
> | SSDRec| 4.52 | 1.95 | 4.19 | 3.28 | 2.03  | 0.72  |
> | DreamMiss | 6.90 |4.34 | 5.48 | 4.77 | 2.65| 0.88 |
>
> From the experimental results, we observe that the performance of STEAM and SSDRec surpasses that of SASRec, but falls short compared to our method, DreamMiss, further validating the effectiveness of DreamMiss. We include these results in Table 1 of our revision.

---

> ### Author Response · Authors · 2024-11-22
> **Response to Reviewer fpPx --- Part 8/10**
>
> ### Weakness 4
> > **Scalability Limitations:** Insufficient discussion of performance on large-scale recommendation systems. Limited **analysis of computational resource requirements** for real-world deployment. Need for more detailed discussion of **potential optimization strategies for larger datasets**.
>
> Thank you for your valuable suggestion. Below we provide a more detailed discussion of DreamMiss's application to larger-scale recommendation systems.
>
> - We first analyze the **computational complexity of DreamMiss** and **discuss its performance on large-scale recommendation systems**.
>   - **Computational complexity:** DreamMiss employs a Transformer as the sequence encoder and a multi-layer perceptron (MLP) as the denoising model. **Training phase:** The Transformer model consists of a Self-Attention Mechanism, a Feed-Forward Neural Network, and Layer Normalization. Let $L$ be the length of the input sequence, $d$ be the embedding dimension, and $N$ be the number of layers in the model. The complexity for computing self-attention at each layer is $O(L^2\cdot d)$, and the complexity for the MLP network is $O(L\cdot d^2)$. For $N$ layers, the overall computational complexity becomes $O(N\cdot (L^2\cdot d+L\cdot d^2))$. **Inference phase:** Let $K$ be the number of reverse steps, the computational complexity of DreamMiss can be $O(K\cdot N\cdot (L^2\cdot d+L\cdot d^2))$.
>   - **Complexity comparison:** The computational complexity of DreamMiss for **training each epoch is nearly similar** to other diffusion-based recommenders and traditional recommenders that **use the same sequence encoder**. Furthermore, since we employ DDIM to accelerate sampling during the inference phase, the value of $K$ for DreamMiss is significantly lower than that of DreamRec. In addition to Table 5 of the appendix of our original submitted manuscript, we compare the time cost per epoch of DreamMiss and other methods. As shown in Table R6. DreamMiss **substantially reduces the time cost during the inference phase** than DreamRec, and **has a similar training time cost with other methods**.
>
>     **Table R6:** Running time comparison of DreamMiss and other methods on three datasets per epoch.
>     | Methods | YooChoose || KuaiRec || Zhihu ||
>     |---------|-----------|-----------|-----------|-----------|-----------|-----------|
>     |         | Training | Inferencing | Training | Inferencing | Training | Inferencing |
>     | SASRec  | 01m 38s   | 00m 06s    | 02m 07s  | 00m 08s     | 00m 10s  | 00m 01s    |
>     | AdaRanker | 02m 29s  | 00m 08s    | 03m 38s  | 00m 09s     | 00m 14s  | 00m 01s    |
>     | DreamRec (DDPM) | 01m 31s | 21m 32s | 03m 59s  | 32m 40s     | 00m 14s  | 01m 31s    |
>     | DreamMiss (DDIM) | 01m 22s | 00m 13s | 02m 23s  | 00m 11s     | 00m 11s  | 00m 01s    |
>
>
>
> - **Computational resource requirements in deployment**. In our experiments, we set the batch size as 256. In this setting, the GPU memory usage is around 2000 MB, which is comparable with SASRec. The time cost for each epoch is presented **in Table R6**. In large-scale recommendation systems, although the number of users and items increases, the sequence encoder (Transformer) and denoising models (MLP) we employ do not require excessive computational resources. Furthermore, Thompson sampling consumes almost no computational resources without involving model parameters. These factors enable our approach to be effectively applied in large-scale systems. Furthermore, optimizing the computational efficiency of Denoising Diffusion Models (DDMs) during both training and inference has become an important research focus. Improvements in this area would enhance the practical use of our proposed approach.

---

> ### Author Response · Authors · 2024-11-22
> **Response to Reviewer fpPx --- Part 9/10**
>
> - Then, we detail some **potential optimization strategies** for larger datasets.
>   - Model Selection [1]: Choose lightweight architectures with lower computational complexity as the sequence encoder.
>   - Accelerating inference for DDMs: Reduce the iterative steps in the inference phase by consistency models [2], knowledge distillation [3], or Reflow [4].
>   - Starting Item [5]: Instead of using pure noise as the starting item, use the mean of the sequence to reduce the iteration times during inference.
>
>
> We include these analyses in Appendix F.7 of our revision.
> [1] Yang Li, Tong Chen, Peng-Fei Zhang, and Hongzhi Yin. Lightweight self-attentive sequential recommendation. In CIKM, pp. 967–977. ACM, 2021a.
>
> [2] Yang Song, Prafulla Dhariwal, Mark Chen, and Ilya Sutskever. Consistency models. In ICML, volume 202 of Proceedings of Machine Learning Research, pp. 32211–32252. PMLR, 2023a.
>
> [3] Chenlin Meng, Robin Rombach, Ruiqi Gao, Diederik P. Kingma, Stefano Ermon, Jonathan Ho, and Tim Salimans. On distillation of guided diffusion models. In CVPR, pp. 14297–14306. IEEE, 2023a.
>
> [4] Xingchao Liu, Chengyue Gong, and Qiang Liu. Flow straight and fast: Learning to generate and transfer data with rectified flow. In ICLR. OpenReview.net, 2023b.
>
> [5] Xiao Lin, Xiaokai Chen, Chenyang Wang, Hantao Shu, Linfeng Song, Biao Li, and Peng Jiang. Discrete conditional diffusion for reranking in recommendation. In WWW (Companion Volume), pp. 161–169. ACM, 2024a.
>
>
> ### Weakness 5
> >**Dataset Limitations:** Need for validation on more diverse and larger-scale datasets. Limited demonstration of effectiveness across different domains and data distributions.
>
> We appreciate your valuable suggestion! To demonstrate the effectiveness of DreamMiss on larger and more diverse datasets, we have included additional experiments on the Steam dataset, one of the largest datasets commonly used in research, as well as on the Amazon-Beauty and Amazon-Toys datasets, which come from different domains.
> - We first introduce the details of the three datasets. The Steam dataset is a collection of data related to the Steam gaming platform, with over three million interactions. Amazon Beauty and Amazon Toys
>   are two categories of datasets from the Amazon platform. They
>   consist of user reviews and ratings for beauty/toy products available on the Amazon website. We treat all ratings as implicit feedback and organize them chronologically by the timestamps. Items and users with fewer than 5 interactions are filtered. The statistics of this dataset are listed in **Table R7**.
>
> **Table R7:**  Statistics of the Steam, Beauty, and Toys datasets.
>
> |             |   Sequence  |     Items    |Ineteractions|
> | ----------- | ----------- | ------------ |------------ |
> |    Steam    |   281,428   |    13,044    | 3,485,022   |
> |   Beauty    |   22,363    |    12,101    |   198,502   |
> |    Toys     |   19,412    |    11,924    |   167,597   |

---

> ### Author Response · Authors · 2024-11-22
> **Response to Reviewer fpPx --- Part 10/10**
>
> - Then we **conduct experiments** to compare the performance of DreamMiss with diverse baselines on the three datasets. Experimental results are presented in **Table R8**. The baseline methods are the same as those in **Table 1 of our original submitted manuscript**.
>
>
> **Table R8:** Overall performance of different methods for the sequential recommendation on diverse datasets (Beauty and Toys) from different domains. The best score and the second-best score are bolded and underlined, respectively.
>
> | Methods       | Steam (H@20 %)       | Steam (N@20 %)       | Toys (H@20 %)       | Toys (N@20 %)       | Beauty (H@20 %)     | Beauty (N@20 %)     |
> | :------------ | :------------------: | :------------------: | :------------------: | :------------------: | :------------------: | :------------------: |
> | GRU4Rec       | 9.23  | 3.56  | 3.18  | 1.27   | 3.85   | 1.38   |
> | Caser         | 15.20  | 6.62   | 8.83  | 4.02  | 8.67   | 4.36  |
> | SASRec        | 13.61 | 5.36  | 9.23   | 4.33   | 8.98 | 3.66   |
> | Bert4Rec      | 12.73 | 5.20   | 4.59   | 1.90  | 5.79   | 2.35   |
> | CL4SRec       | 15.06  | 6.12  | 9.09   | 5.08   | 10.18  | 4.85  |
> | IPS           | 15.65  | 6.46   | 9.29  | 5.27  | 10.15  | 4.56   |
> | AdaRanker     | 15.71 | 6.68   | 8.18   | 4.33   | 8.03   | 3.80   |
> | DiffuASR      | 15.74 | 6.59  | $\underline{9.39} $ | 5.19 | 10.03  | 5.16   |
> | CaDiRe        | 15.65  | 6.42   | 9.33   | 5.16  | 9.85   | 4.46  |
> | PDRec         | $\underline{15.78}$ | 6.51  | 9.08  | 5.12   | 10.24  | $\underline{5.02} $  |
> | DiffRec       | 15.09  | $\underline{6.89} $ | 9.18  | $\underline{5.25}$ | 10.21 | 5.14   |
> | DreamRec      | 15.08 | 6.39   | 9.18   | 5.22  | $\underline{10.32} $ | 4.88   |
> | DreamMiss     | **16.19** | **7.52** | **9.88** | **5.39** | **10.72** | **5.40** |
> | improv.       | 2.53%              | 8.38%              | 4.96%              | 2.60%              | 3.73%              | 7.04%              |
>
> Our method consistently outperforms various baselines on **larger dataset** (Steam) and **diverse datasets** (Amazon-beauty and Amazon-toys), **further highlighting the effectiveness of DreamMiss**. We include additional experimental results in Appendix F.1 of our revision.
>
> We sincerely hope that our additional response has adequately **addressed your concerns**. If so, we would greatly appreciate your consideration in **raising the score**. If there are any remaining concerns, **please let us know**, and we will **continue to actively address your comments** and **improve our work**.

---

> ### Author Response · Authors · 2024-11-25
>
> Dear Reviewer fpPx,
>
> We deeply appreciate your valuable feedback and the time you've taken to review our work with these insights, especially during this busy period.
>
> Regarding your primary concern about the **theoretical justification for the method of simulation**, we have clarified why we simulate missing data instead of compensating for it, supported by additional experimental results. Moreover, we provide a theoretical analysis of our method through extrapolation and consistency regularization. For the point you raised about **validation of Dual-side Thompson Sampling (DTS)**, We have clarified the advantages of DTS and diffusion models. To validate these claims, we have conducted additional experiments to assess the generalizability of DTS to other recommenders and compared different sampling strategies with DTS. Moreover, we have proposed additional metrics to evaluate the reliability of continuity and stability, as well as the robustness of the probability models through experiments. Concerning **more baselines and more datasets**, we have conducted experiments on two additional baselines and expanded our methods to include larger datasets (Steam) as well as more diverse datasets (Beauty and Toys).  For the point about **scalability limitations**, we have analyzed the computational complexity and computational resource requirements of DreamMiss in comparison to other recommenders. Additionally, we have discussed the potential optimization strategies for larger datasets. We have included all these clarifications and experiments in our revision.
>
>
>
> We are reaching out to kindly inquire about the current status of your review regarding our response and revision. We sincerely hope that our responses have adequately addressed your concerns. Furthermore, we are eager to address any additional queries you might have, which will enable us to enhance our work further.
>
> Thank you once again for your support and understanding!
>
> Best regards,
>
> Authors

---

> ### Author Response · Authors · 2024-11-30
> **Kind Reminder**
>
> Dear Reviewer fpPx,
>
> As the discussion phase is coming to an end, we sincerely hope to address your concerns and engage in further discussion. If you have any remaining questions, please feel free to reach out directly. We would like to take this final opportunity to improve our work.
>
> Best regards,
>
> Authors of Paper 7669

---

> ### Author Response · Authors · 2024-12-03
> **Looking forward to your feedback**
>
> Dear Reviewer fpPx,
>
> As the discussion phase is coming to an end, we sincerely hope to address your concerns thoroughly. If you find our responses satisfactory, we would greatly appreciate your consideration of revising your score. Thank you once again for your valuable feedback and thoughtful suggestions.
>
> Best regards,
>
> Authors of Paper 7669

---

### Official Review · Reviewer_jqjS · 2024-11-04

**Soundness:** 3
**Presentation:** 3
**Contribution:** 2
**Rating:** 6
**Confidence:** 2

**Summary:**

The proposed method aims to tackle overlooking the missing data within the guidance for diffusion models. Specifically, they first detect the missing data by constructing local and global models and checking the continuity and stability. Based on the detected missing data, the authors then train diffusion models with uncertainty-aware guidance. Extensive results are provided to support the effectiveness.

**Strengths:**

- This paper is well-structured and easy to follow.
- The proposed method is well-motivated and novel.
- The details of the experiments are revealed, and the code is released, which will ease the reproducibility of this paper.

**Weaknesses:**

- Some typos exist. For example, the first character in line 72 should be capitalized.
- This paper is not well-motivated. Why is Thompson sampling the best choice to derive the guide signal for a diffusion-based recommendation? Besides, no related experiments can verify the best of the Thompson sampling strategy compared to other sampling strategies.
- I noticed that the scale of datasets used in the experiments is relatively small, with interaction counts of fewer than one million. I recommend that the authors conduct experiments on larger datasets to further demonstrate the effectiveness of the proposed method, or the method's efficiency may be questioned.

**Questions:**

All my questions have been included in the weaknesses.

---

> ### Author Response · Authors · 2024-11-22
> **Response to Reviewer jqjS --- Part 1/3**
>
> Dear Reviewer:
>
> We would like to express our sincere gratitude for your valuable feedback and constructive suggestions. We have detailed our responses **point-by-point** to your comments. We sincerely hope that our additional response has adequately **addressed your concerns**.
>
> ### Weakness 1
>
> > **Typo:** The first character in line 72 should be capitalized.
>
> Thanks again for your thorough review and feedback on our paper. Regarding your comment about the capitalization of the first character in line 72, we have carefully examined the context. Since there is a comma preceding that character at the end of page 1, it should indeed remain lowercase, and we believe this is consistent with standard conventions.
> It may be that the formatting of this sentence immediately following Figure 1 led to this misunderstanding, and we will work on improving our clarity in the revision.
>
> ### Weakness 3
>
> >  **Evaluation on larger datasets:** Experiments on larger datasets should be conducted to further demonstrate the effectiveness of the proposed method.
>
> We appreciate your suggestion to further improve our work. To illustrate the effectiveness of DreamMiss on larger datasets, we perform an additional experiment on the `steam` Dataset which is one of the largest datasets commonly used in research.
>   - **Details of the Steam dataset:** The Steam dataset is a collection of data related to the Steam gaming platform, containing over **three million user-game interactions**. we treat all ratings as implicit feedback and organize them chronologically by the timestamps. Items and users with fewer than 5 interactions are filtered. The statistics of this dataset are listed in Table R2.
>
>
>     **Table R2:**  Statistics of the Steam dataset.
>
>     |             |   Sequence  |     Items    |Ineteractions|
>     | ----------- | ----------- | ------------ |------------ |
>     |    Steam    |   281,428   |    13,044    | 3,485,022   |
>
>
>  - **Experimental results:** We compare the performance of DreamMiss with diverse baselines on the Steam in **Table R3**. The baseline methods are the same as those in **Table 1 of our original submitted manuscript**.
>
>     **Table R3:** Overall performance of different methods for the sequential recommendation on Steam. The best score and the second-best score are bolded and underlined, respectively.
>
>     | Methods         | Steam (HR@20 %)  |  Steam (NDCG@20 %) |
>     | :-------------- | :--------------: | :--------------:   |
>     | GRU4Rec         | 9.23  | 3.56     |
>     | Caser           | 15.20  | 6.62     |
>     | SASRec          | 13.61  | 5.36    |
>     | Bert4Rec        | 12.73  | 5.20   |
>     | CL4SRec         | 15.06  | 6.12    |
>     | IPS             | 15.65 | 6.46     |
>     | AdaRanker       | 15.71 | 6.68     |
>     | DiffuASR        | 15.74  | 6.59   |
>     | CaDiRe          | 15.65 | 6.42    |
>     | PDRec           | $\underline{15.78}$ | 6.51    |
>     | DiffRec         | 15.09   | $\underline{6.89}$  |
>     | DreamRec        | 15.08  | 6.39     |
>     | DreamMiss       |**16.19**| **7.52**  |
>     | improv.         | 2.53%            |  8.38%             |
>
>     Based on the experimental results, our method consistently outperforms various baselines on **larger dataset**, **further highlighting the effectiveness of DreamMiss**. **We include additional experimental results in Appendix F.1 of our revision.**

---

> ### Author Response · Authors · 2024-11-22
> **Response to Reviewer jqjS --- Part 2/3**
>
> ### Weakness 2
>
> > **Motivation for Thompson Sampling:** The reason for Thompson sampling as the best choice to derive the guide signal for diffusion-based recommendation is not clear. Related experiments should be conducted to verify the best of the Thompson sampling strategy compared to other sampling strategies.
>
> Thank you for your valuable feedback, which is of great help in improving the clarity of our motivation.
>
>   **Table R1:** The performance comparison of **different sampling strategies**  in uncertainty-aware guided diffusion for recommendation.
>
>   | Methods | YooChoose |  | KuaiRec |  | Zhihu |  |
>   | :---: | :---: | :---: | :---: | :---: | :---: | :---: |
>   |  | H@20(%) | N@20(%) | H@20(%) | N@20(%) | H@20(%) | N@20(%) |
>   | Base |4.78 | 2.23 | 5.16 | 4.11| 2.26 | 0.79 |
>   | Random-based | 6.24 | 3.91 | 5.37 | 4.19 | 2.30 |0.80 |
>   | Time-based| 6.28 | 3.96 | 5.20 | 4.55 | 2.24 |0.79 |
>   | Model-based | 6.49 | 3.62 | 5.18 |4.59 | 2.36 | 0.80 |
>   | DreamMiss | **6.90** |  **4.34** | **5.48** | **4.77** | **2.65** | **0. 88** |
>
>
> - We first **explain why we use the Dual-side Thompson Sampling** to create uncertainty-aware guidance for recommendation, which can **simulate missing data stochastically** while maintaining user preference.
>
>   - **Characteristics of TS:** TS [1] strategy can balance the exploration and exploitation based on the known user preference in probability models. To capture user preference evolution, the probability models are established based on item continuity and sequence stability metrics. The Dual-side Thompson Sampling is **more likely to** remove items with high continuity in sequence with greater stability. Such removal **has little impact on understanding user preference evolution pattern**, as evidenced by the consistent green curves between the observed sequence and the one edited by DTS in Figure 1 of our original submitted manuscript.
>
>   - **Comparison with other sampling strategies:** We compare DTS with different sampling strategies, such as random-based sampling, time-based sampling [2], and model-based sampling [3]. Random sampling samples items uniformly without exploiting known information, making it difficult to capture key patterns. Time-based sampling samples items based on the timestamps of interaction, which may overlook long-term or short-term preference. Model-based sampling samples items based on another trained model, which requires additional training resources and lacks supervision signals. All three strategies may **disrupt users' original preference evolution** when sampling, as the green curves of user preference evolution change in (a), (b), and (c) of Figure 6 of our revision.
>
>
>
> - We then **conduct experiments** to verify the advantages of the TS strategy **compared to other sampling strategies**. The experimental results are presented **in Table R1**. The `Base` represents the diffusion-based recommender without a sampling strategy. The results of `random-based` are from `w/o GL` **in Table 2 of our original submitted manuscript**, which replaces probability models with random sampling. The experimental results indicate that different sampling strategies have a noticeable impact on the diffusion-based recommender compared to the `Base` approach. However, DreamMiss demonstrates superior performance relative to other sampling strategies, **confirming the advantages of DTS**.
>
> - **Clarification:** We propose DTS as a promising solution to address missing data in the diffusion-based sequential recommendation. It offers a **simple yet effective implementation** of the idea to **simulate uncertain missing data while preserving user preferences**, thus achieving DDMs' consistency regularization. While DTS offers distinct advantages, we **may not claim it to be the best sampling strategy**; rather, it is a well-performing strategy compared with other competitive strategies in our task as shown in Table R1.
>
> **We will include the motivation for DTS and experimental results in Appendix B.2 and F.5 of our revision.**

---

> ### Author Response · Authors · 2024-11-22
> **Response to Reviewer jqjS --- Part 3/3**
>
> **References**
>
> [1] Ian Osband and Benjamin Van Roy. Bootstrapped thompson sampling and deep exploration. CoRR, abs/1507.00300, 2015a.
>
> [2] Ehsan K. Ardestani and Jose Renau. ESESC: A fast multicore simulator using time-based sampling. In HPCA, pp. 448–459. IEEE Computer Society, 2013.
>
> [3] Xiaohui Chen, Jiankai Sun, Taiqing Wang, Ruocheng Guo, Li-Ping Liu, and Aonan Zhang. Graph-based model-agnostic data subsampling for recommendation systems. In KDD, pp. 3865–3876. ACM, 2023.
>
>
> We sincerely hope that our additional response has adequately **addressed your concerns**. If so, we would greatly appreciate your consideration in **raising the score**. If there are any remaining concerns, **please let us know**, and we will **continue to actively address your comments** and **improve our work**.

---

> ### Author Response · Authors · 2024-11-25
>
> Dear Reviewer jqjS,
>
> We deeply appreciate your valuable feedback and the time you've taken to review our work with these insights, especially during this busy period.
>
> Regarding your primary concern about the **motivation for TS**, we have clarified the advantages of TS and compared it with other sampling strategies with experiments. For the point you raised about **experiments on larger datasets**, we have conducted additional experiments using the Steam dataset, one of the largest datasets commonly utilized in research. We have included these clarifications and experiments in our revision.
>
> We are reaching out to kindly inquire about the current status of your review regarding our response and revision. We sincerely hope that our responses have adequately addressed your concerns. Furthermore, we are eager to address any additional queries you might have, which will enable us to enhance our work further.
>
> Thank you once again for your support and understanding!
>
> Best regards,
>
> Authors

---

> > ### Comment · Reviewer_jqjS · 2024-11-26
> >
> > Dear Authors,
> >
> > I appreciate the authors for their response. However, I will maintain my original score and ratings.

---

> > > ### Author Response · Authors · 2024-11-28
> > >
> > > Dear Reviewer jqjS,
> > >
> > > We sincerely appreciate your acknowledgment of our efforts and rebuttal.  Your insights on **the motivation for TS** and **more experiments on larger datasets** have significantly helped us in improving our manuscript.
> > >
> > >  If our additional clarifications and experiments have properly addressed your concerns, we kindly ask if you could consider revising the overall rating of our paper. If there are any remaining concerns, we would be more than happy to continue improving our work based on your feedback.
> > >
> > > Once again, thank you for your support and understanding!
> > >
> > > Best regards,
> > >
> > > Authors of Paper 7669

---

### Author Response · Authors · 2024-11-23
**General Response**

Dear Reviewers/ACs/SACs/PCs,

We would like to summarize the strengths of this work acknowledged by reviewers, the contributions of our work, and the revision and responses we have made to address all the reviewers’ concerns.

**Strengths**:
- Motivation and Novelty: well-motivated (`Reviewer jqjS`), novel (`Reviewer jqjS`), the problem of missing data is crucial (`Reviewer fpPx`)

- Presentation: well-structured and easy to follow (`Reviewer jqjS`), generally well-written (`Reviewer fpPx`), overall well-written (`Reviewer VkFH`)

- Experimental effectiveness: ease the reproducibility (`Reviewer jqjS`), experimental results are sound (`Reviewer VkFH`)


**Contributions of our work:**
 - We propose uncertainty-aware guided diffusion for the sequential recommendation, addressing the issue of uncertain missing data.
 - We design a novel Dual-side Thompson Sampling strategy to simulate uncertain missing data, which samples based on two probability models (i.e., item continuity and sequence stability).
 - Theoretical analysis through extrapolation and consistency regularization justifies the effectiveness of DreamMiss, supported by extensive experiments.


**Responses and revision** (highlighted in red text in the PDF) to address these reviewers’ concerns are **summarized** as follows:

- `Reviewer jqjS`:
  - **Motivation for TS:** We have clarified the advantages of TS and compared it with other sampling strategies. Through additional experiments, we have demonstrated the superiority of DTS and claim it as a well-performing implementation of our idea rather than the best sampling strategy. We have included the clarification and experiments in Appendix B.2 and F.5 of our revision.

  - **Experiments on larger datasets:** We have conducted additional experiments using the Steam dataset, one of the largest datasets commonly utilized in research. The results further validate the effectiveness of DreamMiss. We have included the experiments in Appendix F.1 of our revision.


- `Reviewer fpPx`:
   - **Theoretical justification for the method of simulation:** We have clarified why we simulate missing data instead of compensating for it, supported by experimental results comparing three methods: simulating missing data with DreamMiss, using observed sequences with DreamRec, and compensating for missing data with DreamImp. Additionally, we provide a theoretical analysis of our method through extrapolation and consistency regularization. We include the theoretical analysis and experiments in Appendix B.1, B.3, and F.3.
   - **Validation of Dual-side Thompson Sampling (DTS):** We have clarified the advantages of DTS and diffusion models. To validate these claims, we have conducted additional experiments to assess the generalizability of DTS to other recommenders and compared different sampling strategies with DTS. Moreover, we have proposed additional metrics to evaluate the reliability of continuity and stability, as well as the robustness of the probability models through experiments. We have included the explanation and experiments in Sections 4.3 and Appendix B.2, F.4, and F.5 in our revision.
   - **More Baselines and more datasets:** We have conducted experiments on two additional baselines and expanded our methods to include larger datasets (Steam) as well as more diverse datasets (Beauty and Toys). We have included additional experimental results in Table 1 and Appendix F.1 of our revision.
   - **Scalability Limitations:** We have analyzed the computational complexity and computational resource requirements of DreamMiss in comparison to other recommenders. Additionally, we have discussed the potential optimization strategies for larger datasets. We include these analyses in Appendix F.7 of our revision.

- `Reviewer VkFH`:
  - **Clarification for missing data issue:** We have discussed the missing data issue that DTS can address and included the discussion in Appendix F.3 of our revision.
  - **Rationale of accelerated sampling for DDIM:** We have provided a more detailed explanation of the rationale behind DDIM's accelerated sampling and compared it with DDPM. We include it in Appendix C of our revision.
  - **Reasonablility of stability scores:** We have clarified the calculation of stability scores, which are based on item continuity scores within a sequence and align with traditional methods.

We warmly encourage you to review the results in the revised manuscript. Hope our response and additional experiments could address your concerns!

Once again, we deeply appreciate the time and expertise you have shared with us. Your encouraging feedback motivates us to continue advancing this work for the broader community, and we are more than happy to add clarifications to address any additional recommendations and reviews from you！

Best regards,

Authors of Paper 7669

---

### Meta-Review · Area_Chair_UTFN · 2024-12-21

**Metareview:**

The proposed method addresses the challenge of overlooked missing data in the guidance of diffusion models. The approach begins by detecting missing data through the construction of local and global models, assessing continuity and stability to identify gaps. Using the detected missing data, the authors train diffusion models with uncertainty-aware guidance to improve robustness. Extensive experimental results demonstrate the effectiveness of the proposed method.

Positive points:
+ This paper is well-written and organized well.
+ The studied problem is important.

Negative points:
- The model lacks clear theoretical justification (why the model can work)
- The baselines are not sufficient.
- The proposed method can be not scalable.

**Additional Comments On Reviewer Discussion:**

In the rebuttal period, the authors have provided detailed responses to the comments of the reviewers. However, the reviewers do not change their scores. It should be noted that the reviewer who provided a 10 rating is not considered in my evaluation, since she did not provide any meaningful comments. Overall, I think the paper is very borderline, and I slightly tend to reject the paper, since there are no significantly high ratings to support it.

---

### Decision · Program_Chairs · 2025-01-22

Reject